# A High-Flux Compact X-ray Free-Electron Laser for Next-Generation Chip Metrology Needs

James B. Rosenzweig [1,*], Gerard Andonian [1], Ronald Agustsson [2], Petr M. Anisimov [3], Aurora Araujo [2], Fabio Bosco [1], Martina Carillo [4], Enrica Chiadroni [4], Luca Giannessi [5], Zhirong Huang [6], Atsushi Fukasawa [1], Dongsung Kim [3], Sergey Kutsaev [2], Gerard Lawler [1], Zenghai Li [6], Nathan Majernik [6], Pratik Manwani [1], Jared Maxson [7], Janwei Miao [1], Mauro Migliorati [4], Andrea Mostacci [4], Pietro Musumeci [1], Alex Murokh [2], Emilio Nanni [6], Sean O'Tool [1], Luigi Palumbo [4], River Robles [6], Yusuke Sakai [1], Evgenya I. Simakov [3], Madison Singleton [6], Bruno Spataro [5], Jingyi Tang [6], Sami Tantawi [6], Oliver Williams [1], Haoran Xu [3] and Monika Yadav [1]

1  Department of Physics and Astronomy, University of California, Los Angeles, 470 Portola Plaza, Los Angeles, CA 90095, USA; gerard@physics.ucla.edu (G.A.); fuka@g.ucla.edu (A.F.); miao@physics.ucla.edu (J.M.); sean.otool2020@gmail.com (S.O.); obw@physics.ucla.edu (O.W.)
2  RadiaBeam Technologies, 1717 Stewart Ave., Santa Monica, CA 90404, USA; murokh@radiabeam.com (A.M.)
3  Los Alamos National Laboratory, Los Alamos, NM 87545, USA; smirnova@lanl.gov (E.I.S.); haoranxu@lanl.gov (H.X.)
4  Dipartimento di Scienze di Base e Applicate per l'Ingegneria, University of Rome "La Sapienza", 00161 Rome, Italy; enrica.chiadroni@uniroma1.it (E.C.); mauro.migliorati@uniroma1.it (M.M.); luigi.palumbo@uniroma1.it (L.P.)
5  INFN Laboratori Nazionali di Frascati, Via Enrico Fermi, 54, 00044 Rome, Italy; bruno.spataro@lnf.infn.it (B.S.)
6  SLAC National Accelerator Laboratory, 2575 Sand Hill Rd., Menlo Park, CA 94025, USA; zrh@slac.stanford.edu (Z.H.); majernik@slac.stanford.edu (N.M.); nanni@slac.stanford.edu (E.N.); riverr@stanford.edu (R.R.)
7  Department of Physics, Cornell University, 109 Clark Hall, Ithaca, NY 14853, USA
*  Correspondence: rosen@physics.ucla.edu

**Abstract:** Recently, considerable work has been directed at the development of an ultracompact X-ray free-electron laser (UCXFEL) based on emerging techniques in high-field cryogenic acceleration, with attendant dramatic improvements in electron beam brightness and state-of-the-art concepts in beam dynamics, magnetic undulators, and X-ray optics. A full conceptual design of a 1 nm (1.24 keV) UCXFEL with a length and cost over an order of magnitude below current X-ray free-electron lasers (XFELs) has resulted from this effort. This instrument has been developed with an emphasis on permitting exploratory scientific research in a wide variety of fields in a university setting. Concurrently, compact FELs are being vigorously developed for use as instruments to enable next-generation chip manufacturing through use as a high-flux, few nm lithography source. This new role suggests consideration of XFELs to urgently address emerging demands in the semiconductor device sector, as identified by recent national need studies, for new radiation sources aimed at chip manufacturing. Indeed, it has been shown that one may use coherent X-rays to perform 10–20 nm class resolution surveys of macroscopic, cm scale structures such as chips, using ptychographic laminography techniques. As the XFEL is a very promising candidate for realizing such methods, we present here an analysis of the issues and likely solutions associated with extending the UCXFEL to harder X-rays (above 7 keV), much higher fluxes, and increased levels of coherence, as well as methods of applying such a source for ptychographic laminography to microelectronic device measurements. We discuss the development path to move the concept to rapid realization of a transformative XFEL-based application, outlining both FEL and metrology system challenges.

**Keywords:** metrology; semiconductor; ptychography; laminography; free-electron laser; photoinjector; beam dynamics; undulator; coherence; X-ray

## 1. Introduction

We examine in this paper the exciting potential application of the X-ray free-electron laser (XFEL) for meeting the urgent and extremely demanding needs of next-generation semiconductor chip metrology. The XFEL [1] is a transformative instrument [2,3] that for a decade and a half has produced unprecedented capabilities in imaging due to its ultrafast timescale of illumination, wavelength (Angstrom), and, above all, high degree of coherence. These properties permit the use of rapidly evolving coherent diffraction imaging (CDI) techniques [4,5] with femtosecond time resolution at Angstrom wavelength. Combined, these attributes allow so-called 4D imaging of atomic and molecular systems [6], which evolve at these spatio-temporal scales. With hard X-rays, the penetrating nature of the radiation further permits imaging of dense condensed matter systems. With such new horizons opened up, the XFEL [7–9] has strongly impacted the recent trajectory of a wide swath of science, spanning an impressive range of applications in nanoscience from ultrafast chemistry [10], atomic physics [11], and molecular dynamics [12] and on to high-energy-density physics [13], among many other scientific areas [14].

When considering the XFEL for translational applications (e.g., imaging of industrial nanomaterials [15]) such as our current focus, the imaging of next-generation computer chips, there are numerous practical and fundamental problems to overcome. First, on the practical side, despite a boom in deployment of new XFELs in national laboratories worldwide, there remains an acute shortage of beam time. This may be mitigated by the introduction of much more compact and less costly ways of realizing XFEL capabilities. Addressing the challenges of "miniaturizing" the XFEL footprint and price tag is currently an extremely active research area. It entails a rethinking of the underlying physical methods for more effective beam creation and acceleration, advanced magnetic systems, new measurement capabilities, and innovations in X-ray optics. The current work arises from one such research effort, termed the ultracompact X-ray free-electron laser (UCXFEL) [16], which is fundamentally enabled by use of the newly emerging technique of very-high-field cryogenic radio-frequency (RF) acceleration.

It is well appreciated in the photon science community that for imaging applications in industry and medicine, the dimension of the object under study using single-shot CDI methods based on the use of XFELs is limited by the spatial coherence and spot size of the X-ray beam. As such, only small objects (a few tens of μm) can be imaged in this way. To image extended objects, all the way to the macroscopic scale, one may employ a scanning CDI method termed ptychography, where the coherent X-ray beam is scanned across the object with some overlap in neighboring illuminated spots. This allows one to reconstruct the macroscopic (up to the cm size, relevant to semiconductor chips) object's structural details with nanometer-scale resolution. Finally, we note that for objects such as chips where three-dimensional information is needed, the penetrating nature of hard (Angstrom-class) X-rays is essential. By illuminating the macroscopic object in a 2D ptychographic scan and then rotating it in the appropriate way, one may pair ptychography with laminography [17], which is a technique related to tomography optimized for near-planar objects.

First using a coherent photon source based on a modern X-ray light-source storage ring, the burgeoning techniques of ptychographic tomography and laminography have been validated for such applications. These proof-of-principle experiments performed at existing national lab-scale facilities are quite convincing in the physical results obtained, with dramatic 3D reconstructions obtained, as discussed below in detail. Solutions based on large user facilities such as a ring light source, which are optimized for shared use from a wide scientific community, are not applicable for an industrial instrument dedicated to chip manufacturing and inspection processes due to cost and size considerations. Further, the sources used thus far in proof of principle are inadequate in the average coherent flux obtained, making measurement times impractically long for industrial uses. Alternatives developed in compact laboratory environments also do not produce sufficient flux for this current application [18]. Here, we discuss in some detail a promising solution for moving

forward with an XFEL for chip metrology based on extensions to hard X-ray operation of a recently developed, rapidly maturing model for a soft X-ray UCXFEL.

We begin discussion of the proposed FEL by providing some needed background. The compact XFEL light source based on advanced, high-gradient accelerator technologies has attracted considerable recent attention [19–21]. The compact XFEL research thrust, which with its new accelerator methodology is termed a fifth-generation light source, promises to radically reduce the cost and footprint of present X-ray FELs. Current XFELs require space and investment at the km/USD billion level and are practical only for applications in national labs as a result. One of the first initiatives to take on the fifth-generation light-source challenge was the DARPA-sponsored AXIS program, which sought to harness dielectric laser accelerators (DLAs), advanced electron sources, and microundulators to create a highly compact XFEL [22,23]. This instrument was intended to address the particular need for new medical imaging techniques, with an eye towards eventual field deployment for military use. The AXIS program, while highly successful, was cut short by a shift in research funding priorities. The UCLA–Stanford–RadiaBeam-centered GALAXIE collaboration within the AXIS program (which forms an essential component of the present initiative) has been comprehensively involved in new directions in fifth-generation light sources traceable to the AXIS program.

AXIS and GALAXIE gave way to many subsequent high-impact new research directions. GALAXIE produced a serious push towards the development of microundulators and other MEMS-based beam optics devices. AXIS also gave rise to the accelerator-on-a-chip program (ACHIP) initiative [24], aimed at the development of compact dielectric laser accelerators, and the ultracompact X-ray free-electron laser program. The UCXFEL initiative has identified in detail a pathway to beams able to use short-period undulators, with a solution utilizing both very-high-brightness beam creation and acceleration uniquely enabled by cryogenic high-gradient RF cavities. With short-period undulators, one may obtain an X-ray free-electron laser operating at a 1 nm wavelength with a footprint and cost (<40 m including X-ray beamlines; <USD 35 million) that permit distributed coherent X-ray FEL sources for research, medical, industrial, and security applications, as discussed below in further detail. The proposed layout of this system is shown in Figure 1.

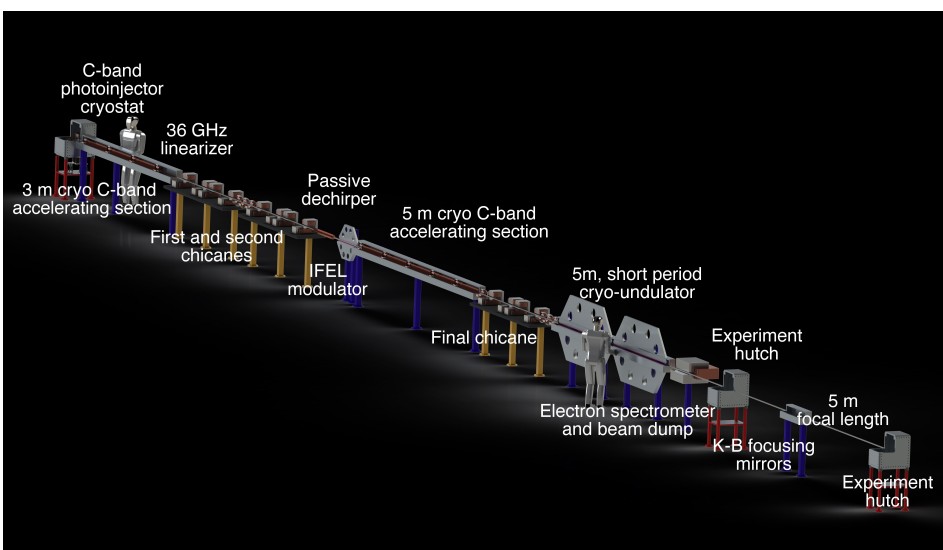

**Figure 1.** The layout of the soft X-ray UCXFEL (from Ref. [16]).

While the science and applications case is compelling in the soft X-ray regime and is currently being pursued, the new, very-high-impact applications in semiconductor metrology have emerged as urgent national needs. Looking to the future, next-generation semiconductor chip technology may be impacted through UCXFEL-based metrology by enabling powerful, nondestructive imaging techniques. The challenge will be to create a

compact XFEL system design for ptychographic tomography aimed at real-time–minute-level–semiconductor metrology at few-nanometer resolution, well beyond the current state of the art.

This proposed hard X-ray, high coherent-flux UCXFEL system is presented with a clear vision as to how it may be implemented, with several component systems currently at the frontier of the relevant technologies. However, the system we propose here is one in which many such challenging systems must be integrated to produce a reliable, complete system which creates and uses, in the most intricate way, coherent hard X-rays to yield a new, essential tool for the chip manufacturing industry. This requires a combined examination of the XFEL and ptychographic laminography systems. Similarly, FELs in the EUV-to-soft-X-ray wavelength level are now being vigorously pursued to enable next-generation chip lithography sources. The research and development for a very-high-flux, highly coherent XFEL represents a comparable level of challenge, the details of which we describe in the following sections.

## 2. The Ultracompact X-ray Free-Electron Laser: Background and Status

The ultracompact X-ray free-electron laser, as indicated above, was driven by several innovations arising in fifth-generation light-source research. The first was the promise of short-period undulators driven by MEMS technology, in which one searched for solutions to the compact XFEL with period length $\lambda_u \simeq 100$ μm. With such a short $\lambda_u$, the value of the undulator strength $K_u = \frac{eB_0\lambda_u}{2\pi m_e c}$ for attainable field strength $B_0 \leq 1.5$ T does not contribute notably to the resonant relation for the on-axis radiation wavelength $\lambda_r$

$$\lambda_r = \frac{\lambda_u}{2\gamma^2}\left[1 + \frac{1}{2}K_u^2\right]. \tag{1}$$

In this case, particularly for hard X-ray operation, a scaling law for the XFEL gain (or Pierce) parameter $\rho$ is given approximately by

$$\rho \simeq \frac{\gamma^6}{3}\left(\frac{\lambda_r}{2\pi}\right)^4 B_{6D}. \tag{2}$$

The parameter is important to FEL performance in a number of ways, the most prominent being through the gain length

$$L_g = \frac{\lambda_u}{4\pi\sqrt{3}\rho}; \tag{3}$$

The factor $B_{6D}$ in Equation (2) is the six-dimensional beam brightness, or density in 6D phase space, usually written as

$$B_{6D} = \frac{2I}{\varepsilon_x\varepsilon_y\sigma_{\delta p/p}}. \tag{4}$$

In making the XFEL more compact through the use of a smaller-period undulator, one directly benefits by the lower beam energy needed, as the linac is a main contributor to the instrument's length and cost. However, Equation (2) indicates that a lowered energy ($U = \gamma m_e c^2$) strongly diminishes $\rho \propto \gamma^6 \propto \lambda_U^2$. To utilize a modest advance in undulator period, at least an order of magnitude increase in $B_{6D}$ would be needed to compensate the loss in $\rho$. One may point out that in terms of undulator length, the factor of $\lambda_u$ serves to automatically shorten the gain length, and thus, we expect that the undulator length needed (about $20L_g$ for SASE saturation) shrinks more quickly than the energy needed. However, the solution to obtaining increased $B_{6D}$ demands the use of higher-gradient accelerating fields, and this also serves to push the physical length of the linac down further.

With the discovery that one may operate a copper linac structure at cryogenic temperatures with significantly lower losses and much higher breakdown fields [25], the problem

of the UCXFEL design became practical. This remarkable improvement comes about due to several physical mechanisms: (1) the surface resistivity of the copper is lowered by a factor of up to 4.8 (in C-band); (2) the thermal conductivity shares the heat dissipated due to this resistivity more effectively at low temperature; (3) the coefficient of thermal expansion is greatly diminished, thus strongly mitigating the stress on the material; and (4) the material yield strength is much higher, and the electric-field-driven breakdown is strongly suppressed. Indeed, at 40 degrees Kelvin, the maximum surface field becomes 500 MV/m, without noticeable breakdown. The practical field, limited by the onset of strong dark current emission, is slightly above 300 MV/m. For our designs in both the RF photoinjector and linac sections, we limit the peak on-axis field to at or below 250 MV/m.

This discovery impacts first the electron source. It has been found that $B_{6D}$ scales with the applied electric field at injection as $\sim E_0^{2.5}$. With this enhancement in hand, one predicts an increase in 6D brightness by a factor of $\sim 15$ over the present state of the art. In familiar terms, for a 100 pC beam injected with full-length 5 psec (20 A peak current), one obtains a final normalized emittance after compensation of $\varepsilon_n \simeq 50$ nm-rad. This beam must be compressed using a two-stage chicane-based system which includes a laser-induced microbunching (inverse free-electron laser, IFEL) interaction to bring the beam to kA level without significant emittance growth. A list of parameters describing this design is given in Table 1.

**Table 1.** Laser modulator and chicane parameters for the IFEL compression system. Any unrepresented parameter is the same value as in Ref. [16].

| Parameter | Units | 1 µm Value | 3 µm Value |
|---|---|---|---|
| Modulator peak magnetic field | T | 0.265 | 0.477 |
| Modulator laser peak power | MW | 200 | 80 |
| Modulator laser waist size | mm | 0.5 | 0.5 |
| Final chicane bend angle | degrees | 0.855 | 1.66 |

With such a bright beam available for use, a design for a 1 nm wavelength ultracompact X-ray FEL has been enabled. It uses a similar peak field in the linear accelerator sections (linacs), which (along with the photoinjector) are of a new, independent coupling design with a very reentrant, high-shunt-impedance cavity shape. These linacs are only 8 m in active length and yield a 1 GeV beam, which lases at the desired 1 nm wavelength using a $\lambda_u = 6.5$ mm high-field cryogenic permanent magnet undulator. This undulator has a narrow gap, which may be strongly affected by resistive wall losses, leading to a loss of gain. Fortunately, the IFEL compression-induced microbunching serves to mitigate this effect.

With such strong acceleration in place, new scenarios for beam collective effects come into consideration. The effects of intrabeam scattering become more prominent in increasing the slice energy spread due to the high beam density and low initial longitudinal temperature. Short-range wakefields in combination with space charge and strong RF-induced focusing present challenges in short-range beam breakup (SR-BBU) control, and simulation tools to study these effects and effective countermeasures have been developed [26]. For the present application, we will need, as discussed below, to use multiple beam pulses per RF fill, and this gives rise to consideration of long-range transverse beam breakup (LR-BBU). Extensions of the simulation tools developed for SR-BBU are used to address these issues.

This design is concentrated on compactness and cost control (the instrument including soft X-ray optics is <40 m in length and <USD 40 million in cost), producing a design that pushes the state of the art in many ways. As the aim in this case was to place such instruments with modest flux (a few % of a full-scale FEL) in a university environment, such concerns are of high importance. In the present industrial context, however, these issues are less urgent. A more risk-averse technical design approach is thus considered here. It will introduce, however, new challenges in obtaining harder X-rays, higher repetition rates, higher overall X-ray flux, and improved transverse coherence, as demanded by ptychographic laminography techniques.

The development work performed in the three years since the publication of the 1 nm UCXFEL design is considerable in scope. It has concentrated on obtaining a basic understanding of cryogenic RF performance in C-band, as opposed to the original experimental work performed in X-band (8–12 GHz), where beam dynamics issues are more challenging. Recent work has included dedicated experimental development on the cryo-RF gun, cryogenic photoemission, general C-band (4–8 GHz) techniques, and wakefield management. We describe this progress in what follows, concentrating first on the C-band cryogenic photoinjector work and related RF and cryogenic technology research.

### 3. Recent Cryogenic Photoinjector Development

The high-brightness RF photoinjector is a key element in the design approach we adopted for UCXFEL design. As such, much emphasis has been placed on cryogenic C-band RF photoinjector research. The core of the recent development of C-band cryogenic RF methods has focused on the establishment of the basics of high-gradient-structure physics, design, and testing. The near-term goal is to create and test the performance of the high-brightness, very-high-field (up to 250 MV/m on the photocathode) photoinjector gun. The existing approach to the UCXFEL high-brightness photoinjector has been extensively discussed in previous publications. Here, we adopt the basic design principles described before, particularly in Ref. [27]. From the standpoints of optimization of the single-bunch beam dynamics and the tolerance to high-gradient operation [16,27,28], the performance of the cavity and the beam upstream of the first linac section are essentially unchanged.

To implement this approach practically, more mature, detailed considerations are then necessary in order to enable the feasible realization of a cryogenic high-brightness 1.6-cell gun. A prototype photoinjector cryostat cutaway design is shown in Figure 2. Multifaceted challenges are presented in this experimental scenario that span a wide range of technological and basic physics research and development. Without loss of generality, three important topics to consider here are the effects of temperature on cathode photoemission; the feasibility of enveloping cryogenic in situ gun tuning and alignment; and the scaling of infrastructure costs, particularly in moving to higher-repetition-rate operation.

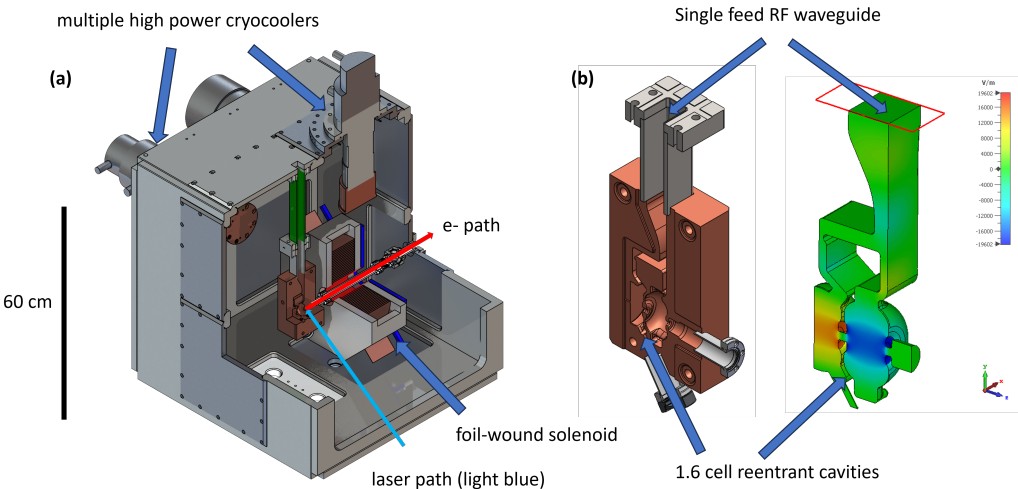

**Figure 2.** (**a**) Cutaway of design for 1.6-cell photoinjector gun section showing relative locations of laser penetration, beam path, and basic cryogenic hardware. (**b**) UCLA-SLAC-LANL cryo-RF gun design alongside cavity-mode fields from simulation.

In order to meaningfully address these three areas experimentally, a CrYogenic Brightness-Optimized Radiofrequency Gun (CYBORG) facility was developed and commissioned at the UCLA MOTHRA Lab. This facility features a first-generation cryogenic RF gun: a C-band $\frac{1}{2}$-cell reentrant design which shows the advantages of such cavity shapes, as proposed for use with distributed coupling in the $\pi$ mode [29]. This is the mode which is necessary to obtain the desired beam dynamics in the eventual multiple-cell gun design.

At the MOTHRA laboratory, CYBORG and its associated beamline further act as a development platform for cryogenic RF components and beamline integration [30]. The gun operates at more modest fields and temperatures than the eventual 1.6-cell cryogenic photoinjector design and so represents an important stepping stone. In the second, longer-term phase of development, CYBORG will function as a cryogenic, high-gradient test bed for novel photocathode measurements, with a coupled load lock for cathode plug exchange. A photograph of the current state of phase 1 CYBORG along with a schematic cutaway of phase 2 configured for the INFN-style minipuck cathode testing are shown in Figure 3. The initial testing of CYBORG is concentrated on cryo-cooler-enabled 77 K operation, due to both near-term practicality and to the potential impact on implementation in the UCXFEL.

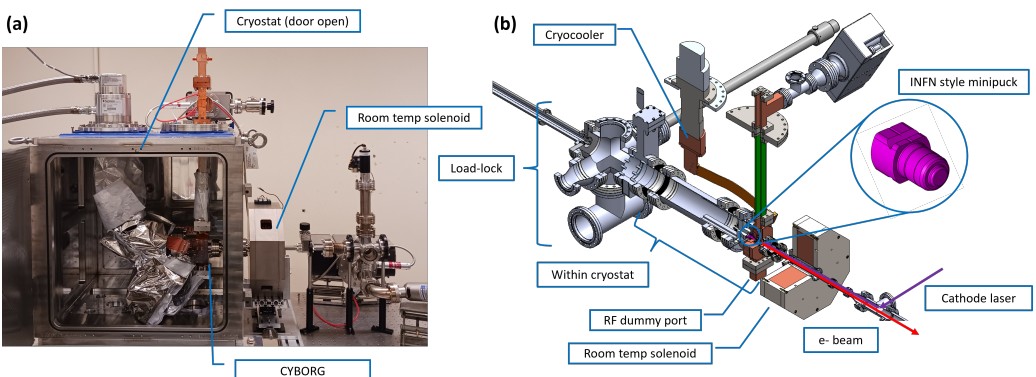

**Figure 3.** CYBORG beamline development in two configurations: (**a**) existing phase 1 setup (using Cu cathode) with cryostat door open, and (**b**) phase 2 configuration cutaway with load lock for INFN-style minipuck cathodes.

The MOTHRA lab is also the site of several additional ongoing experiments investigating basic cryogenic C-band physics at both low and high power [31,32]. Perhaps of highest relevance so far is the initial exploration of the *Q*-enhancement available as a function of temperature for C-band RF in copper. Measurements have found that the *Q*-enhancements at the two proposed operating temperatures for the UCXFEL and $C^3$ projects are 4.61 at 45 K and 2.9 at 77 K, respectively [33]. These enhancements are consistent with the variability predicted by models of the anomalous skin effect which dominates at cryogenic temperature. This is essential input for proceeding with the new UCXFEL design.

The highly complex physics of novel cryogenic cathodes is of deep relevance to the performance of a photoinjector operated simultaneously at very low emittance and at high average current. The state-of-the-art performance of cathodes with low mean transverse energy (MTE) and high quantum efficiency (QE) in cryogenic operation is of high interest, especially in the case of near-threshold photoemission. For our application here, a possible order-of-magnitude reduction discussed previously in MTE is not quite possible, but nonetheless, notable improvements may be in reach [28]. Indeed, the reduction in MTE by 70–80 percent should still be expected for metallic cathodes operated at cryogenic temperatures as predicted by existing theoretical treatments [34]. We calculate these values analytically for a range of values in excess energy and temperature for copper cathodes in Figure 4 to illustrate generally expected metallic cathode behavior. We note that operating at threshold leads to asymptotic linear scaling between MTE and temperature, so for a 45 °K working point, we would expect a factor-of-seven reduction.

The improvements from a theoretical cryogenic metallic cathode come at the expense of a strong decrease in quantum efficiency, so in this application, the possibility of using semiconductor cathodes (e.g., $Cs_2Te$) with significantly higher QE is more relevant. In an extremely high-gradient cavity, this will greatly increase beam current and consequent beam brightness for given laser parameters. At CYBORG, this scenario can be explored up to just above 120 MV/m peak field on the cathode. Higher fields are envisioned for a

room-temperature gun being developed for the CARIE Facility at LANL [35–37], which is of similar RF design as the 1.6-cell cryogenic gun described next.

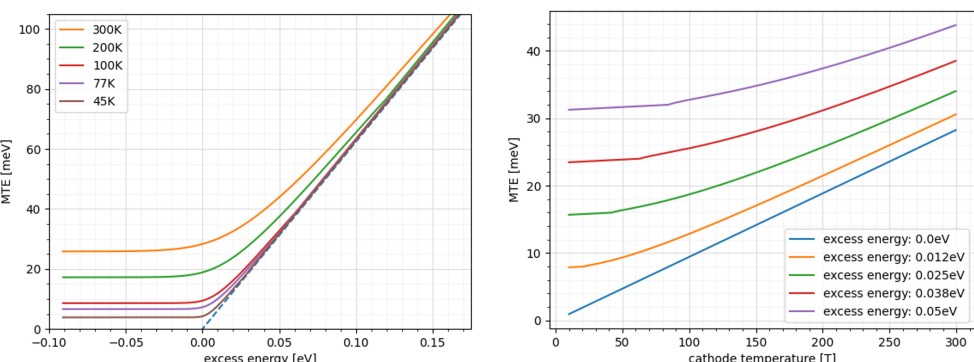

**Figure 4.** Near-threshold photoemission calculations and the effect on mean transverse energy (MTE) for copper cathodes in certain ranges relevant to the UCXFEL photoinjector. Impact of laser photon tuning on MTE in cryogenic limit. We expect to obtain a factor-of-seven reduction from 45 K operation (**left**). Temperature dependence on MTE in near-threshold limit. For 0 excess energy, note the asymptotic linear scaling of T and MTE (**right**).

The lessons learned at MOTHRA in the context of CYBORG heavily informed the photoinjector design shown in Figure 2. The gun RF geometry and cavity couplers were designed in a joint UCLA-SLAC-LANL collaboration. One additional feature to note is the presence of a cryogenic multistart foil-wound solenoid within the cryostat, designed to reduce brightness, reducing magnetic field aberrations. The 1.6-cell cryogenic-ready cavities are in the process of fabrication now, and the next-generation cryostat and the foil-wound solenoid are in advanced states of development. This design has been developed using physical and engineering principles established at the MOTHRA lab using the CYBORG beamline. The design parameters mirror, as noted above, existing UCXFEL values including >240 MV/m accelerating fields; while the original UCXFEL gun design envisions a 45 K operational temperature with an attendant $Q$-enhancement of 4.6, operation at 77 K with a more modest increase in $Q$ is under consideration for the UCXFEL presented in this paper.

A new style of multistart, foil-wound solenoid has been developed which can improve the performance of the cryogun by removing damaging field components. Beamline solenoids are typically wound helically or as "pancakes", where the wire is wound radially in before crossing over and winding out. Both of these approaches break rotational symmetry and introduce higher-order multipole moments which can degrade beam emittance. Multistart foil winding suppresses selected multipoles, mitigating this problem [38]. Further, since the solenoid must be installed in the cryostat, its thermal performance is also important to consider [27]. This new style of solenoid has notable advantages over conventional designs for conduction cooling, especially considering that flowing liquid cryogens through wires for cooling would be a significant engineering challenge. It is worth considering the possibility of using high-temperature superconducting foils for this application [39].

## 4. The Challenge of Ptychographic Laminography

With this recent background discussed, we now turn our attention to a compelling new application space for compact XFEL technology—advanced semiconductor chip metrology. The issue of chip metrology has recently come to the fore, with a series of reports from the National Institute of Standards and Technology (NIST) [40–42] identifying it as a key national priority in support of the CHIPS Act of 2021. This priority has been assigned due to the urgent needs identified for new technological and scientific methods in support of next-generation semiconductor manufacturing, which in turn set the stated goals of CHIPS

Act-funded and -directed research and development. The first statement in the executive summary of [41] is telling in its clarity:

*"Metrology underpins our ability to address the challenges faced by semiconductor manufacturers. Making investments in metrology capabilities today will future-proof technology needs and support U.S. leadership for the next generation of microelectronics."*

X-ray FEL-based imaging [4] is indeed an attractive candidate for extending metrology applications to future generations of microelectronics. The challenge of measurement through the imaging of integrated circuits to determine quality control in manufacturing, as well as inspection of chips of external provenance for undesired artifacts, requires an enormous range of spatial scales, from a few nanometers to the cm level. At present, one must use destructive, time-consuming techniques, which utilize a variety of approaches from optical microscopy at the meso- to microscopic scale [43] to electron microscopy for nm resolution. To obtain nondestructive information across this variety of resolution scales, recent efforts have shown that coherent X-ray-based 3D ptychographic laminography (PyXL) can be profitably used. This method can zoom through spatial scales, allowing the use of complete chips as samples and exploring the different resolution scales as desired.

The ptychographic laminography scheme and the elaborate setup developed for its demonstration [44] are summarized in Figure 5. This geometry permits 3D imaging of extended objects by ptychographic scanning at each rotational position, collecting thousands of rastered coherent diffraction images. This is then followed by rotation at typically hundreds of rotational positions. The canting of the sample (chip) at a constant angle in laminography is preferred over standard tomography, as it permits a constant thickness of the sample to be traversed by the coherent X-rays. For the X-ray beam to penetrate well the current chip architectures, we should use hard X-rays, preferably over 7 keV in energy.

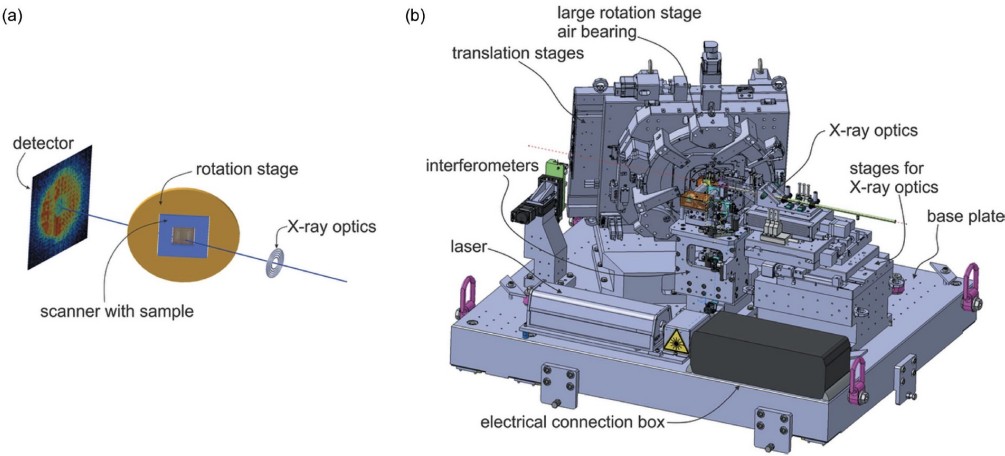

**Figure 5.** (**a**) Schematic of the laminography geometry. (**b**) Overview of the LamNI instrument, from [44].

The initial tests [45,46] of coherent X-ray-based PyXL were carried out at the Swiss Light Source, where a dedicated apparatus was created [44] to perform the needed scanning and rotation. The results have had a resounding impact on the field [46]. These tests, which represent an improvement on previous experiments using a tomographic instead of laminographic technique, showed detailed features and manufacturing flaws of then-current-generation chips at less than 20 nm resolution. Full reconstructions of macroscopic regions of the chips were obtained, allowing fly-through zoom inspection of the imaged structures.

One may perform low-resolution overview scans by choosing different operating conditions and choose a region of interest to examine at higher resolution. This approach permits more efficient and faster inspection methodology. This tiered zoom technique has proven useful in recent scans (see Figure 6), which are performed maximally at ~20 nm

resolution level. The highest-resolution measurements, over limited, partial volumes of the chip, require tens of hours to accomplish.

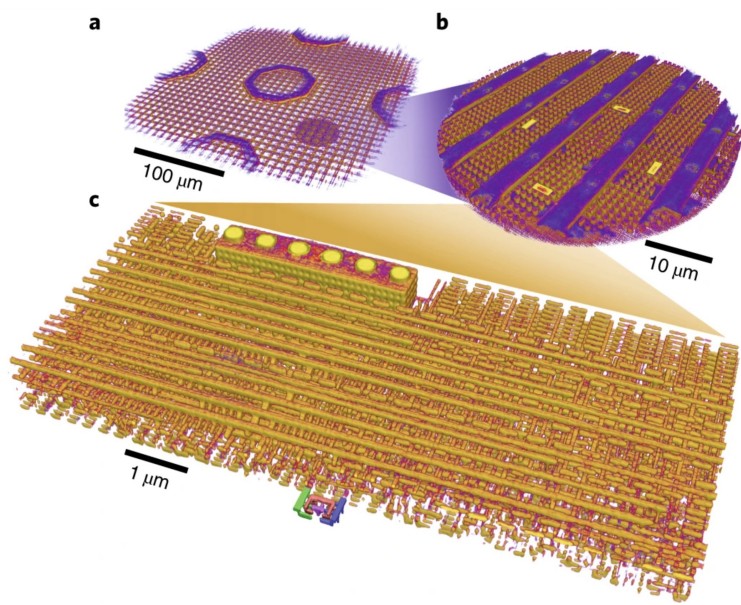

**Figure 6.** Three-dimensional rendering of a ptychographic laminography measurement dataset. (**a**) A combined rendering of the low-resolution and high-resolution datasets. (**b**) A rendering of only the volume measured at high resolution. (**c**) A rendering of a subvolume of (**b**), from Holler et al. [46].

Thus far, the imaging rate, obtainable resolution, and sample size have been limited by the available coherent X-ray flux. For the Swiss Light Source cSAXS beamline, this is $\sim 8 \times 10^8$ 6.2 keV photons per second. In Refs. [46,47], the path to improvements in PyXL measurements based on increased coherent flux was discussed. It was pointed out that the proof-of-principle measurements in [46] were strictly limited in their performance by available flux. Indeed, the prospects for the future based on a $10^4$ increase in flux based on the progression to the use of next-generation ring sources (i.e., the Swiss Light Source upgrade, with an initial factor-of-40 increase in coherent flux) and then moving to an XFEL, are discussed, with the positive implications for resolution increase and/or significant lowering of the time involved in the measurement. As the PyXL approach is dominated by the pixelation of the ptychography, the improvement of resolution with an increase by a factor of $>10^5$, as we find in the UCXFEL, implies an increase in resolution from 19 to 3 nm in repeating the measurements of [46] in a time that is a factor of 300 times shorter.

In predicting future performance for a chip-oriented ptychographic laminography system, one should keep track not only of the total coherent flux but also of the average spectral brightness of the source. Here, we consider an X-ray free-electron laser, so some changes in approach to the mode of XFEL operation beyond self-amplified spontaneous emission are needed to increase this brightness. As is discussed in what follows, we propose a design which can reach coherent hard X-ray flux at high brightness that is an improvement on past PyXL experiments by a factor of over $10^5$. As is noted in [46], the original Si(111) monochromator-based system at the cSAXS beamline sacrifices an excessive amount of flux for creating coherence beyond what is needed for PyXL. Here, we search for a more relaxed solution, which produces high flux at the necessary coherence level.

We use the UCXFEL design as a point of departure. There are other alternative approaches which might be considered, such as the one found in the recent initiative at Arizona State, in which a low-flux coherent soft X-ray FEL-like source is to be obtained based on emittance exchange in low-energy prebunched beams [48,49]; while this option has the advantage of compactness, it suffers from a dramatic ($>10^4$) reduction in available flux and is not extendable to hard X-ray operation. In concentrating on a UCXFEL-based

design for our current application, the XFEL per se is changed in two important ways: first, to obtain hard (7 keV) X-rays, we must increase the beam energy nearly two and a half times to 2.44 GeV; second, FEL design is extended to a regenerative amplifier (RAFEL) scheme based on a periodic train of electron pulses. This so-termed XRAFEL serves to greatly increase the X-ray flux and coherence [50].

We note that the transition from ring light source to XFEL is, from the viewpoint of enhanced coherent flux for the ptychographic laminography imaging of chips, very compelling. This transition of source, as well as related techniques, has not been developed as yet, however, as the dedicated beamline time is not currently available. The lack of opportunity to use XFELs for this important application points out the need for the proposed UCXFEL, which can open the door to implementation of PyXL on a moderate-cost, dedicated instrument.

This article will therefore concentrate mainly on describing this advancement in the coherent X-ray source. However, to evaluate the progress made in scan time, resolution, and sample size, many other factors must be considered. These include management of sample damage; temporal limitations due to sample manipulation, ptychographic scan, and laminographic rotation speed; improvement in algorithms for the reconstruction analysis [51]; and computational resources optimized for this application. We will return briefly to these issues after presentation of the ultracompact hard X-ray FEL design inspired by this compelling application.

## 5. Extending the UCXFEL to High-Flux Hard X-ray, High-Coherence Operation

In order to create a design which may be deployed in a timely fashion, i.e., in the next five years, for next-generation chip inspection purposes, we follow two guidelines: first, to maximally utilize the research performed in support of UCXFEL in the last few years; and second, to adopt a more conservative design strategy than that chosen to make the system as compact as possible.

What remains the same are the choices of RF frequency, cryogenic ultrahigh-brightness photoinjector, and sub-cm period undulator. Further, despite the changes in beam final energy and FEL wavelength, we choose to preserve the philosophy of two-stage pulse compression [52] but with the final compression stage performed by a shorter wavelength IFEL process. The C-band RF frequency is seen as the best current choice for both advanced X-ray FELs and for other applications such as inverse Compton scattering sources and linear colliders. As such, considerable benefit may be anticipated in synergy with the parallel research underway in these areas. The relatively short undulator period (6.5 mm, as in Ref. [16], also based on cryogenic technology) serves to keep the beam energy needed to a minimal level of ∼2.44 GeV for a 7 keV X-ray operation. With this choice, we are again reliant on the very-high-brightness beam that permits high gain and efficiency in the XFEL.

The RF photoinjector needs to be operated in multiple-bunch mode in order to enable an XFEL design philosophy which obtains higher coherence and flux, as described below. The linear accelerator design has evolved to a new, more efficient and elegant form, and this, along with the need for multiple electron bunches per RF pulse, changes our analysis of beam stability. Further, in the interest of near-term feasibility, we propose the option to operate the linacs at room temperature. These choices drive the need to examine further issues, as discussed below.

*Linear Accelerator Design Evolution*

Here, we describe the evolution of the linear accelerator design in the last three years since the original UCXFEL proposal. This evolution is connected to the recent developments in the cool copper collider ($C^3$) initiative [53] and represents a significant step forward in effective implementation of the design concepts introduced in the UCXFEL design [16]. We note in this regard that the chip-metrology-oriented XFEL shares the need for multiple-pulse operation with $C^3$.

Unlike linacs designed for colliders, high-gradient normal conducting linacs for XFELs such as those considered here have smaller charge and nearly negligible beam loading. The power consumption of the linac is thus governed only by the shunt impedance and the desired operating gradient. Recently, a new class of linear accelerators, the distributed-coupling accelerator structure, has been proposed and adopted for the original UCXFEL design, as well as for other applications. This new class of linacs has the advantage of a very high shunt impedance and thus power efficiency. It also has a unique method of construction that reduces the cost of production dramatically in comparison to conventional linacs; this is due to the fact that the number of parts required for conventional linacs is proportional to the number of cavities, while for this new class of linacs, the body of the linac is constructed from only two machined blocks, with a few ancillary components such as couplers and windows. This class of linacs has been high-power-tested both at room temperature and at liquid nitrogen temperature and has achieved record, usable gradients of over 150 MeV/m.

The primary motivation for the invention of this linac topology was to allow for the optimization of the cavity shape, reducing both surface electric and magnetic fields. This resulted in the simplest possible implementation that comprised two manifolds to distribute the RF power uniformly into a set of uncoupled cavities with a Bloch $\pi$-phase advance. This implementation suffers, however, from some undesirable features. First, the structure is essentially a standing-wave accelerator structure that produces a reflection during the filling time that interacts with the RF source. To eliminate the adverse effect of this reflection, a sophisticated low-level RF system is needed. Alternatively, two sections of the structure must be fed together with an RF hybrid to direct the reflected power to a load. This latter solution requires an extra $\lambda/4$ space between the two structures. Such a construction for the GRIT ICS source [54] linac is shown in Figure 7, together with the result of a cold test showing very small insertion losses, on the order of $-35$ dB.

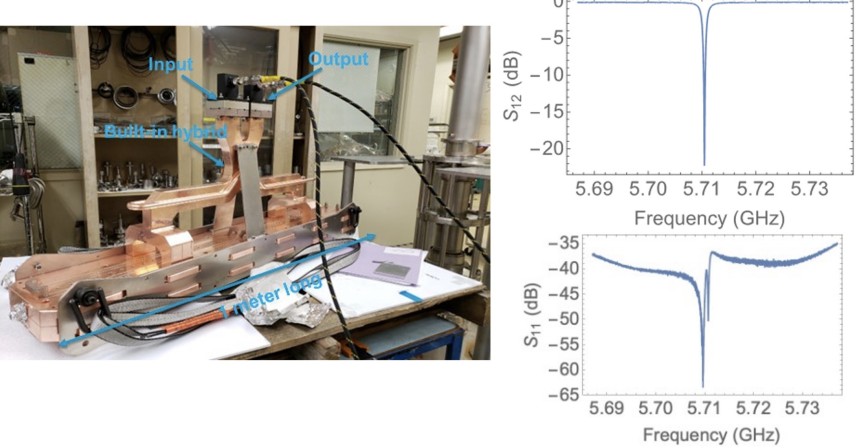

**Figure 7.** The GRIT project linac, an example of a two-section $\pi$-mode structure fed together by a hybrid to eliminate reflections to the source.

The second undesirable feature of the previous design is that the distribution manifold connects to the cavities through a rather long, convoluted waveguide because the guided wavelength in the manifold is longer than the RF free-space wavelength, in the period of which the manifold taps feed to adjacent cavities, as is shown in Figure 8.

The length of this waveguide has to be chosen carefully so that an incorrect operation in one of the cavities, e.g., a detuned cavity due to construction or to a breakdown event, does not result in destroying the uniformity of the power feed to the other cavities with excess power to some of them. Finally, the $\pi$ mode of operation does not result in optimal acceleration of the beam. Indeed, many decades ago, the optimal phase advance to optimize the shunt impedance was studied, and for the traveling-wave geometry at the SLAC linac,

120 degrees was found to be an optimized choice. It can be shown that by optimizing the accelerating spatial harmonic given the reentrant cavity design, the optimum phase advance per period for the present geometry is near 140 degrees.

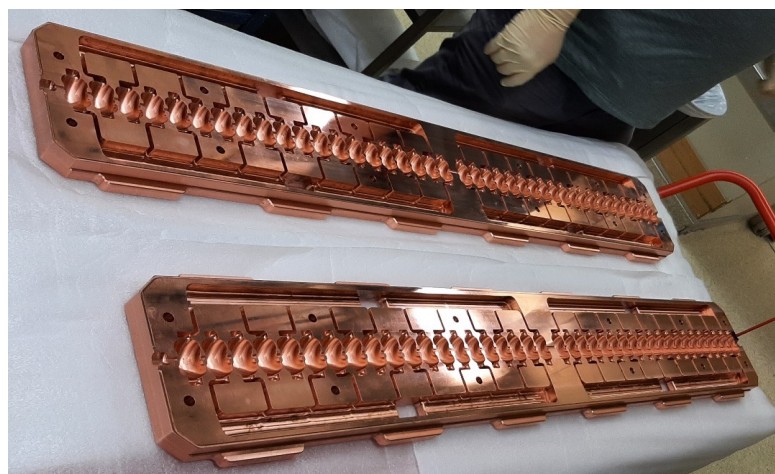

**Figure 8.** The $\pi$-phase-advance distributed-coupling structure before joining, showing power distribution and cavity topology.

For the distributed-coupling linac topology, the choice of the phase advance is arbitrary and can be set to accommodate the demands of efficient acceleration as well as ease of implementation. The initial realization at $\pi$ requires two manifolds, with the guided wavelength within the manifold chosen as $\lambda_g = \lambda_0/2$. The next logical choice would be a phase advance of $2\pi/3$, which requires three manifolds and also the condition $\lambda_g = \lambda_0$, which can only be achieved using a coaxial line or a corrugated waveguide; both options are awkward to implement. The next likely choice, which we believe to be optimal for this topology, is the $3\pi/4$-phase advance. It requires four manifolds with $\lambda_g = 3\lambda_0/2$, which is easily achievable with a smooth rectangular waveguide. It has, as noted above, near to the peak in shunt impedance and by implication a lower surface magnetic field. Figure 9 shows how the shunt impedance for both the best $\pi$ mode and the $3\pi/4$ mode varies as a function of the beam-hole aperture diameter, as evaluated from the optimization of C-band cavities ($f_{RF} = 5.712$ GHz). An example of each optimized cavity shape is shown in Figure 10. The cavities are shown together with the field distribution along the surface of the cavity. Because of the optimized shape, the magnetic field is rather uniform along the cavity surface, and the peak electric field is limited to only twice the accelerating gradient, which is a constraint that was imposed on the optimization process. Notice that the peak magnetic field of the $3\pi/4$ mode is lower than that of the $\pi$ mode, which implies better performance at high gradients. The full assembly including four manifolds for this design is illustrated in Figure 11.

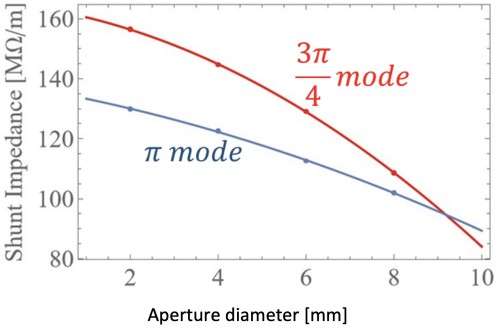

**Figure 9.** Comparison of the shunt impedance for a set of optimized cavities at C-band (5.712 GHz). The cavities are optimized at each aperture parameter for each operating mode.

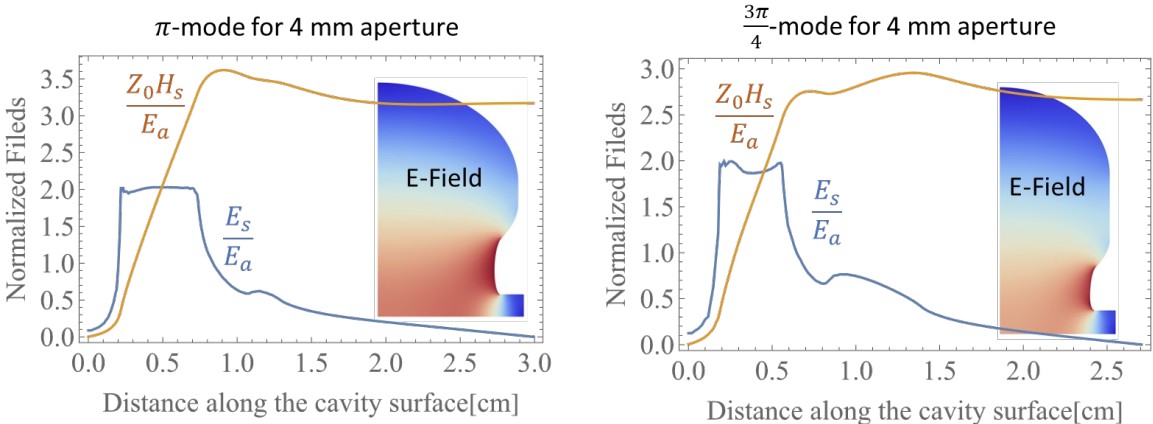

**Figure 10.** Field distribution of optimized cavities for both the $\pi$ mode and the $3\pi/4$ mode. The plotted quantities are surface electric field $E_s$ normalized to the accelerating field $E_a$ and the surface magnetic field multiplied by the free-space wave impedance $Z_0$ and and also normalized to $E_a$.

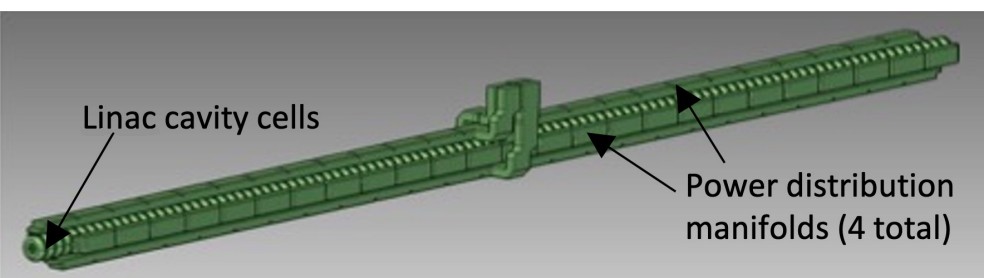

**Figure 11.** Rendering of exterior of $3\pi/4$-phase-advance linac structure and its compact power distribution system. Four waveguide manifolds that transport power to the independently coupled cavities are shown along the sides of the structure.

For the $3\pi/4$ mode, the filling time of a critically coupled cavity is about 330 ns; for the $\pi$ mode, it is near 380 ns. The shorter filling time and higher shunt impedance make the $3\pi/4$ operation more favorable for use in a pulsed, high-gradient linac. Because of the relatively long filling time, it is natural to use a pulse compressor to quickly fill the cavity by overpowering it and using overcoupled cavities. This helps in shortening the total pulse length and hence enables high-gradient operation. Even at this high shunt-impedance level, for operation at an accelerating gradient near 100 MV/m, we would need to supply about 71.5 MW/m, assuming a full beam aperture of about 5 mm. This power level is higher than state-of-the-art klystrons in C-band, assuming they power one-meter structures. This also implies that one should consider the use of pulse compression to obtain the necessary power level. If we dropped the gradient to 80 MV/m, and, hence, the required power level dropped to 46 MW/m, we could operate without pulse compression with one klystron per meter. In this case, pulse compression then could be used to reduce the number of klystrons in the system by a factor of three at the least.

The choice of the operating aperture is dictated by the short-range wakefields for a given bunch charge and length, an issue examined below. If one is operating with high bunch charges (typical in a linear collider) such that the aperture is required to be larger than 8 mm, then one would argue that the linac frequency choice needs to be reduced. With the parameter chosen for the linac needed for the UCXFEL being discussed here, an aperture between 3 and 6 mm with C-band operation should be adequate.

To summarize, to optimize the new distributed-coupling topology, a $3\pi/4$-phase-advance linac design is introduced. This new topology, indeed, remedies all the undesired features of the $\pi$-mode topology. The new structure has four manifolds instead of two, and they are combined with each other in such a manner that the reflection during the filling time is directed to a load. This structure is a standing-wave accelerator that behaves like a

traveling-wave device. The distance between the cavity feed and the power manifold is minimized and is limited only by practical constraints. This is achievable because every manifold feeds every fourth cavity, with a spacing of three halves of a wavelength, which can be matched by a manifold with a guided wavelength of three times that of a free-space wavelength. We note that the construction of this topology is more sophisticated than the earlier incarnation in the $\pi$-mode structure. It requires four layers of copper blocks which must be joined into the highly compact structure.

The development of this type of accelerator structure for ultrahigh-gradient linacs not only is applicable for realizing compact XFEL but also may be naturally applied to a future linear collider [53] such as $C^3$. Its use provides many positive impacts to the UCXFEL design, and these will be discussed in what follows.

## 6. High-Gradient Accelerating Cavity Testing

As discussed above, C-band copper distributed-coupling cavities, with a particular emphasis on cryo-cooled operation, represent a new approach to building the compact accelerating section for the proposed UCXFEL. Using distributed coupling provides numerous degrees of freedom to optimize the cavity geometry to achieve a high gradient and can allow for further increases in accelerating gradients while maintaining the RF power requirements within realistic limits. Building on the demonstration of the dramatically increased breakdown limits shown in X-band testing by the SLAC-UCLA collaboration [25], operation of a cryo-cooled copper distributed-coupling accelerating structure was recently demonstrated also at X-band [55].

To proceed with this design of the hard X-ray UCXFEL, one must evaluate the issues associated with possible cryo-operation of a linear accelerator with the design as outlined above. To this end, a two-cell structure was designed for high-power breakdown testing. Photos of the fabricated first version of the structure are shown in Figure 12. We show here that the structure is fabricated in two halves before being joined via brazing. The structure will also serve as a template for future novel cavity-joining techniques such as diffusion bonding using a paradigm currently under investigation at UCLA.

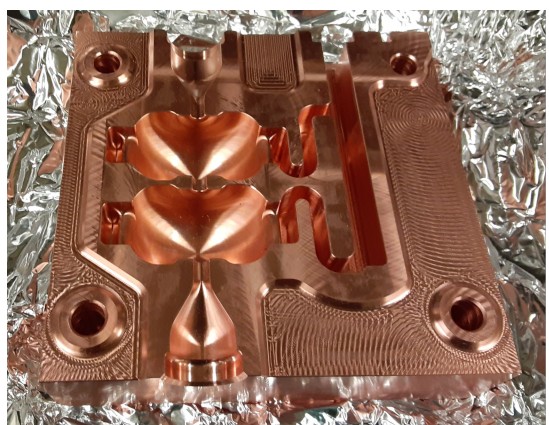 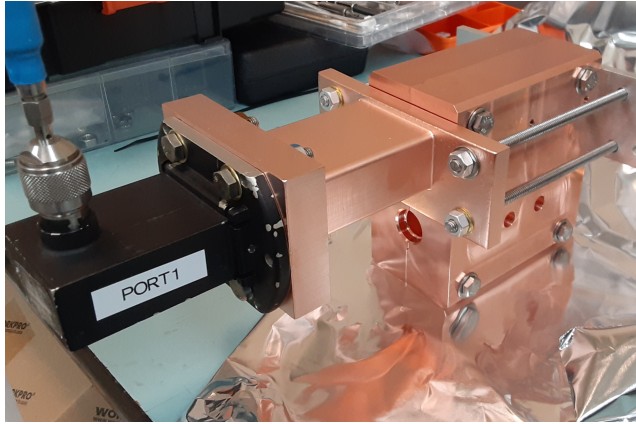

**Figure 12.** Photos of 2-cell brazed structure for high-power breakdown. Photo on the left shows the structure machined in 2 halves and to the right to be brazed into configuration.

To understand the performance envelope of the proposed new $3\pi/4$-phase-advance structure in the most direct way, one should proceed as soon as possible to test the achievable fields in a full one-meter structure. Such structures exist at room temperature at the RadiaBeam GRIT facility [54]. The available peak power at this facility permits operation up to a ∼60 MeV/m acceleration gradient; a planned addition to the RF system of a SLED pulse [56] compressor should permit gradients of 120 MeV/m to be explored.

Should these novel structure geometries prove to maintain needed gradients, for their use, higher-order-mode (HOM) suppression must be studied. Moreover, operation at cryogenic temperatures calls for looking into alternative materials that would provide adequate

HOM damping at low temperatures while being compatible with fabrication processes for distributed-coupling structures. We note that the quadrant structure shown in Figure 13a is fully consistent with the new generation of high-gradient structure fabrication approaches.

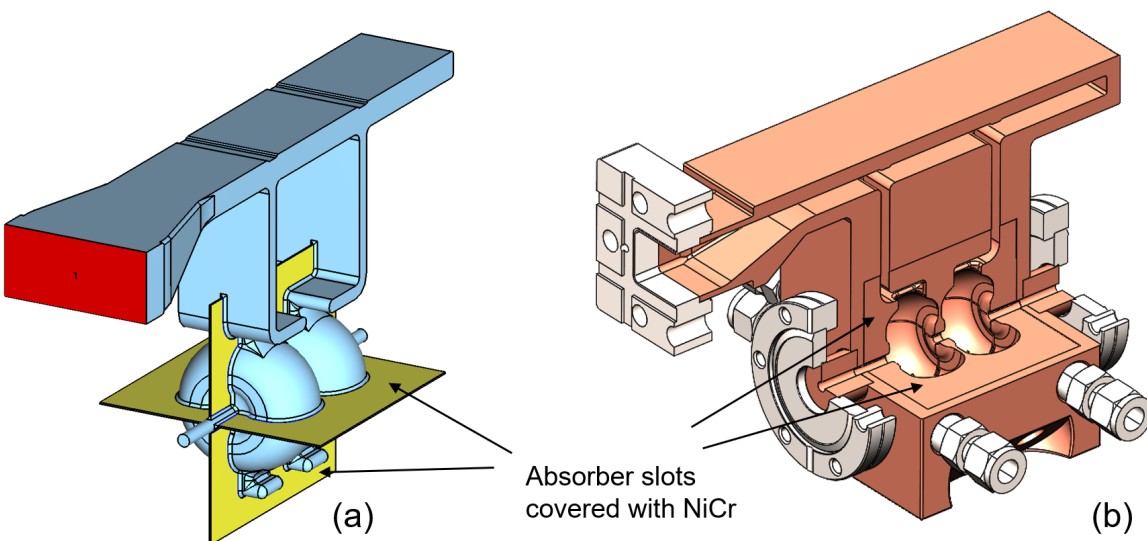

**Figure 13.** The two-cell cavity with NiCr HOM absorbers designed for high-gradient testing: (**a**) CST Microwave Studio model; (**b**) CAD model for fabrication.

A novel design of the HOM couplers was proposed in Ref. [57], which is made with four parallel manifolds coated with nickel–chrome (NiCr) alloy. The authors of Ref. [57] conducted extensive computational investigation and computed the *Q* factors and kick factors for all HOMs in the frequency range from 5 to 40 GHz. It was concluded that NiCr coating reduced ohmic *Q* factors for HOMs and could be used for HOM suppression in combination with appropriate detuning. A procedure for fabricating a NiCr layer in the slots was proposed that involved plating seven alternating layers of 700 nm of nickel followed by 200 nm of chrome, which would later be annealed at a high temperature during the brazing cycle.

To verify the proposed fabrication procedure and to evaluate performance of the NiCr absorbers during high-gradient conditioning of the cavity, a simple two-cell test cavity was designed based on the structures currently under test by UCLA. The design is illustrated in Figure 13. The structure consists of a WR187 waveguide that splits into two smaller waveguides, providing distributed coupling into two RF cavities that operate in the $TM_{01}$ mode with a 180-degree phase advance. The cavity will be fabricated in four quadrants. In the first fabrication step, the quadrants will be machined with the HOM slots, and the layers of Ni and Cr will be deposited into the slots. In the second machining step, the cavity shapes will be machined, removing the NiCr coating everywhere except in the regions highlighted in Figure 13. The four quadrants will then be brazed together to form a single structure, and the nickel and chrome will be annealed.

The fabricated cavity will be tested at the C-band Engineering Research Facility in New Mexico (CERF-NM) at Los Alamos National Laboratory (LANL) [58,59]. The CERF-NM is powered by a 50 MW, 5.712 GHz Canon klystron that produces 50 MW pulses with a pulse length between 300 ns and 1 µs and a repetition rate up to 200 Hz, and it is tunable within the frequency band of 5.707 GHz to 5.717 GHz. The detailed procedure for the high-gradient cavity conditioning and breakdown rates mapping at CERF-NM is outlined in [60]. The cavity will be conditioned to the accelerating gradient of at least 100 MeV/m.

The post-test measurements are similar for the cavities with and without (Figure 13) absorbers in that the cavity will then be cut apart and inspected for damage sustained during the conditioning process. In the quadrant version, damage to NiCr absorbing layers can be expected. The first kind of damage is the damage due to pulse heating that is

expected in the region close to the accelerating cavities. Although the pulse heating of NiCr is expected to not exceed 13 degrees C at the gradient of 100 MeV/m, NiCr has lower thermal conductivity than copper and therefore may not be amenable to pulse heating even at this relatively low magnitude. The second kind of damage is the damage due to HOMs that will be generated during RF breakdowns that will occur in the process of conditioning. The HOMs will penetrate into the damping slots and will be deposited into the NiCr material. Breakdowns are relatively rare events and are not expected to initiate in the vast majority of RF pulses coupled into the cavity. However, multi-MW peak power will be deposited onto the absorbers during the breakdown event. We believe that the proposed tests will serve to qualify NiCr as a new material (with other candidates possible) for HOM absorption in high-gradient linacs with distributed coupling.

## 7. Injection into the $3\pi/4$ Booster Linac and BBU Effects

The original UCXFEL design discussed in Refs. [16,27] assumes the use of distributed-coupling linac structures working with a $\pi$-phase advance. The introduction of the $3\pi/4$-mode structures changes the spatial harmonic content of the on-axis $E_z$ field profile and, thus, the radial focusing coefficient of such structures $\eta$ [27], which is $\eta = 1.12 - 0.5\cos(2\Delta\phi)$ for the $\pi$ mode and $\eta = 0.61 - 0.1\cos(2\Delta\phi)$ for the $3\pi/4$ mode with $\Delta\phi$ being the phase deviation from the crest. As a consequence, the beam dynamics must be reexamined in comparison with the previous results obtained in Ref. [27], with special attention given to matching the beam produced by the high-gradient cryogenic RF gun, see Figure 14 (top), to the first booster linac section. Indeed, a propagation mode, known as *invariant envelope* [61], allows for reducing the projected normalized emittance from 92 nm-rad to 45 nm-rad, see Figure 14 (bottom), by optimized control [61] of the beam's transverse plasma oscillations.

The nominal working point for this propagation mode requires the injection of the beam at its waist ($\sigma_x = \sigma_0$ and $\sigma_x' = 0$) and the adjustment of the average accelerating gradient $E_0$ as a function of the beam current, energy, and spot size [62]. Given these considerations, here, we investigate the dynamics in the first booster linac section following the RF gun in order to identify the working point that minimizes the final emittance. This investigation requires a large number of simulations, so we performed a preliminary scan of the parameter space using the tracking code MILES [26] instead of General Particle Tracer (GPT) [63]. The former is based on simplified, semianalytical models for the description of the self-induced electromagnetic field and thus is particularly convenient for a fast evaluation of such effects. In particular, the space-charge forces are described by a superposition of the fields produced by uniform cylindrical slices, an approach inherited and generalized from the code HOMDYN [64,65] that successfully describes this scenario.

Figure 15 shows the results of the preliminary optimization process with MILES. Here, we kept the high-gradient cryogenic RF gun settings found in Ref. [27] and adjusted the injection distance (the drift $d$ in Figure 15a is measured starting at 112 cm from the cathode) and the average accelerating gradient in the booster linac $E_0$. The global minimum for the final normalized emittance, here 55 nm-rad, was obtained for $d = 5.2$ cm and $E_0 = 73.7$ MV/m, and the evolution of the transverse rms parameters is shown for such values in Figure 15b. We note that although the accelerating gradient is similar to the $\pi$-mode case, this minimum corresponds to a small shift of the injection location downstream of the beam waist, resulting in a slightly larger spot size at the booster exit.

The optimized emittance compensation working point for the new $3\pi/4$-mode standing-wave linac obtained using MILES was validated with detailed particle simulations using the General Particle Tracer (GPT) code with 350K macroparticles. Figure 16 compares the slice emittance information of a 100 pC bunch at the exit of the first booster linac section to that at the entrance for the two modes discussed here. The final emittance found in these simulations is quite similar to the $\pi$-mode case, given that the phase and acceleration gradient in the new linac is flexible—similar performance can be achieved. The projected normalized emittance found in these more detailed simulations is yet smaller than that found in the MILES, at 45 nm-rad. The slice values remain below the 45 nm-rad level in

the body of the beam. The final value of the emittance needed for the FEL design has been established to be near 75 nm-rad; thus, significant performance overhead is obtained in the current design. This overhead may be needed later as a guarantee of emittance dilution during transport and compression, as discussed below.

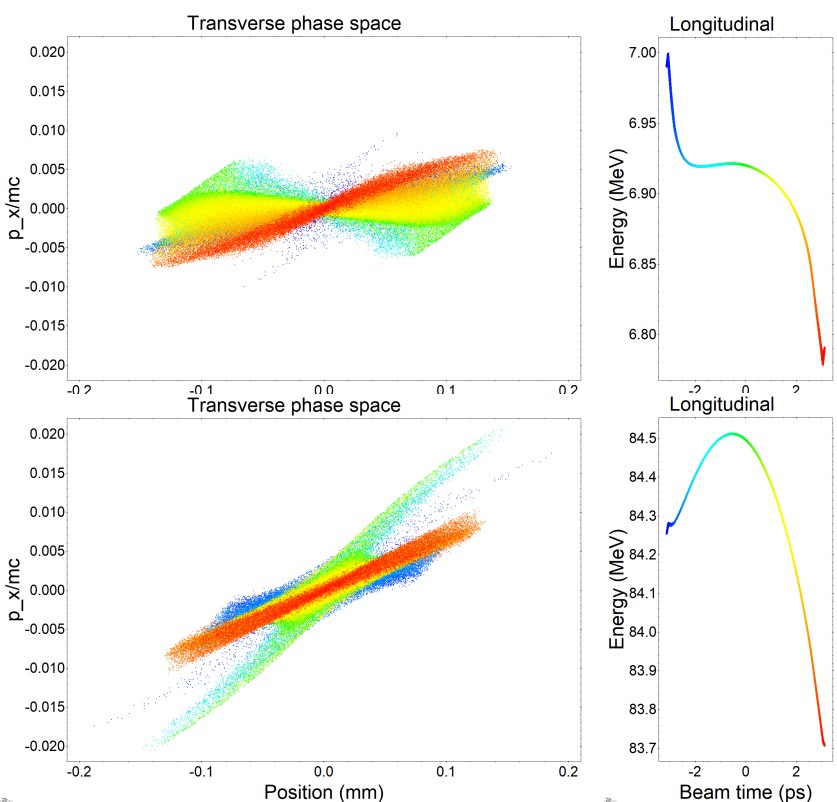

**Figure 14.** Phase spaces at the entrance to (**top**) and at the exit of (**bottom**) the booster linac section colored by the arrival time. One can see that head (blue) and tail (red) slices of the transverse phase space become aligned with core slices (yellow/green) at the exit of the booster linac section. The profile of the final longitudinal phase space suggests near-crest acceleration with 73.8 MV/m in a 105 cm section.

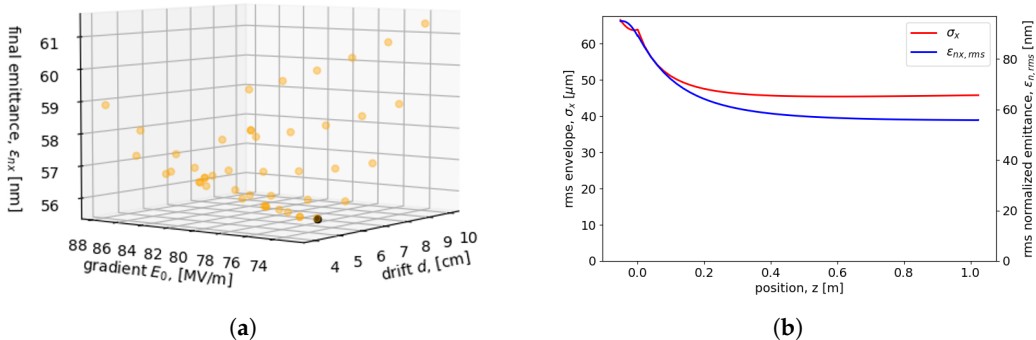

(**a**)                                                                      (**b**)

**Figure 15.** Optimization of the booster linac section working point with on-crest acceleration. The minimum final emittance (black dot) is found over multiple MILES simulations where the drift length and the average accelerating gradient are changed. The distance $d$ is measured starting 112 cm after the cathode, and the gradient is defined as an energy gain per meter of the linac section. (**a**) Global minimum of the final emittance for our parameter space; (**b**) optimal RMS transverse envelope and normalized emittance evolution in the booster linac.

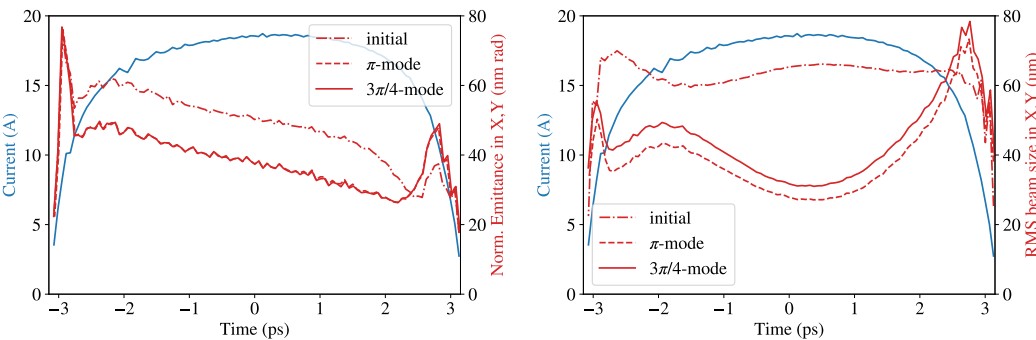

**Figure 16.** Slice information of a 100 pC beam from GPT simulations that shows 45 nm-rad projected normalized emittance performance for $3\pi/4$-mode and $\pi$-mode booster linacs vs. initial 92 nm-rad projected normalized emittance (**left**). RMS beam size of the slices that shows slightly weaker focusing in $3\pi/4$ mode (**right**).

*SRWF-Induced Emittance Dilution*

As the photoinjector is predicted to produce extremely low emittances with relatively high current beams, we must be prepared to preserve this beam quality in order to take advantage of its brightness for the FEL applications. Therefore, in this section, we use MILES to investigate the potential emittance dilution induced by the short-range wakefields (SRWFs) in the high (shunt- and beam-) impedance linac structures themselves, as well as possible mitigation schemes. This SRWF analysis is different from those undertaken in the past [26], due to the foreseen use of the $3\pi/4$-mode linac.

The SRWF interaction for short bunches propagating in periodic accelerating structures is well described by use of diffraction theory [66–68], and asymptotic expressions for the corresponding wake functions are known [69]. A primary concern is represented by the strong dipole wake effects expected for small-cell irises and responsible for a slice-dependent deflection that increases the projected rms emittance. This effect is exacerbated by the low beam rigidity in the initial injection region, which makes particles very sensitive to alignment imperfections. Therefore, mitigation schemes suppressing the BBU effects in the low-energy section are crucial. Here, we present a possible correction technique aimed at finding the trajectory that minimizes the excitation of dipole SRWFs.

It is important to stress that the emittance dynamics in the first booster linac section are very rich and complex: the nominal emittance compensation process and the counteracting dilution induced by the dipole wakes take place simultaneously. It is, thus, extremely helpful to have efficient computational methods capable of including both aspects. The test we perform assumes two sections of a $3\pi/4$-mode distributed-coupling linac: the first uses the parameters obtained from the optimization of the injection working point, while the second simply boosts the energy up to 150 MeV. We also assume that the first section has a 100 μm transverse offset so that the beam propagates in an off-axis trajectory, exciting dipole wakefields in the process.

In Figure 17, the red triangles represent the center of mass trajectory and the rms emittance in this configuration. For comparison, the green squares represent the same quantities without the alignment error. It can be observed that the final emittance is magnified by a factor of ∼2.7 due to the linac offset. An efficient approach to mitigate this effect consists of the introduction of two steering magnets upstream of each linac section that allow for the fine control of both offsets and angles at the injection. Therefore, by using a simple matrix optics calculation [26], one can find the strength of the magnetic correctors that ensures zero offset and angle in each accelerating structure. In such a configuration, the beam is forced to propagate on-axis in each linac section regardless of the misalignments so that the dipole wakefields are completely suppressed and the emittance can be preserved, as shown by the dashed black curves.

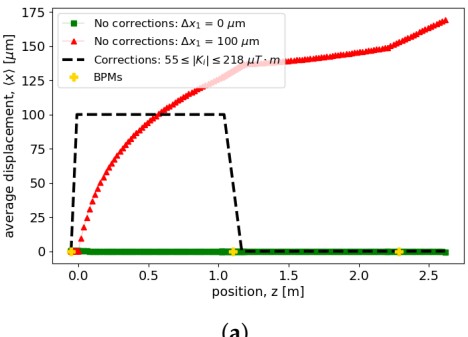 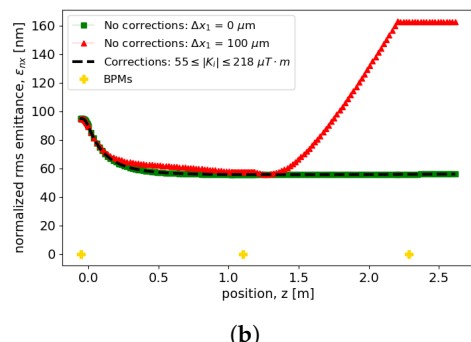

(**a**)     (**b**)

**Figure 17.** Use of trajectory steering for the mitigation of SRWFs. The green squares and red triangles represent, respectively, an ideal machine with no misalignments and a linac whose first section is 100 µm off-axis. Compensation of such an error is achieved by forcing the beam to propagate always on-axis (black dashed lines). The gold crosses identify the location of the BPMs. (**a**) Center of mass trajectory; (**b**) normalized rms emittance.

## 8. IFEL Modulation and Beam Compression

We now move to a discussion of the beam dynamics downstream of the photoinjector. For clarity, we first summarize the key features of the original UCXFEL beamline design [16] on which our new design is based. As shown in the previous section, the cryogenic photoinjector yields 100 pC, 150 MeV beams of 20 A peak current, and ~50 nm-rad normalized projected emittance. For optimal lasing performance, we need to accelerate the beam to a much higher energy—2.44 GeV for the present scenario—with much higher peak current—4 kA. To do so while preserving the extremely high beam brightness we utilize two compression stages. First, the beam is accelerated to 400 MeV energy while accruing a linear chirp that is compressed in a zigzag-style chicane array to compensate for coherent synchrotron radiation (CSR)-induced emittance growth. The residual chirp after compression is removed using a round corrugated dechirping cavity, after which the beam copropagates with an external IR laser in a strong magnetic wiggler to induce a periodic energy modulation. In the original design, that energy modulation had a period of 10 µm. The modulated beam is then accelerated to its final energy, 1 GeV in the original design and 2.44 GeV in this revised design, and compressed in a compact chicane. The energy-modulated longitudinal phase space is converted into a train of 4 kA peak current microbunches whose emittance has been preserved to the 75 nm-rad level thanks to the optimized compression architecture.

As we hinted in the last paragraph, the most notable changes we must introduce with respect to the nominal 1 GeV soft X-ray UCXFEL design are the increased final beam energy of 2.44 GeV and the possibility of changing the period of the IFEL modulation [70]. Lasing at much higher photon energy in this case, nearly 7 keV, dramatically reduces the slippage effects that mandated a 10 µm modulation period in the soft X-ray UCXFEL design. With a shorter modulation period, the momentum compaction required to compress the microbunches is smaller, which helps to compensate for the opposite momentum compaction scaling induced by moving to higher beam energy. In both the beam dynamics simulations and the FEL performance simulations, we consider two options for the modulating wavelength: 1 micron and 3 microns. A 1-micron modulating laser has the smallest required momentum compaction but may start to suffer from slippage effects in the FEL. The 3-micron case requires a slightly stronger compressor, increasing the risk of beam phase-space quality degradation, but reduces slippage effects.

The beam dynamics for this updated working point are essentially unchanged from the original UCXFEL design up to the entrance of the laser modulator. A slightly lower accelerating gradient—100 MeV/m instead of 120 MeV/m—requires three 1 m long C-band linacs rather than two to reach 400 MeV beam energy. This extra meter of acceleration has a negligible impact on the dynamics otherwise. Similar are the compression to 400 A in

the first bunch compressor and subsequent linearization in a round dechirping cavity [71], as was reported in Ref. [16]. The key changes begin in the laser modulator, where, as mentioned previously, we now consider 1- and 3-micron modulating wavelengths. To mitigate changes from the original robust UCXFEL design, we assume the modulator to have the same magnetic period of 15 cm, with the only required change then being in the magnetic field strength to match to the shorter wavelengths. A shorter resonant wavelength in the modulator requires even smaller magnetic fields, so no issues are expected from this change.

The laser modulator is followed by 20 one-meter linac sections to bring the beam to its final energy of 2.44 GeV. As with the modulator, the basic design of the final compressor is unchanged: 10 cm bend magnets arranged with one-meter drifts separating them. The modified design parameters for the two modulating laser working points are given in Table 1. The longitudinal phase space of the final compressed beam is shown for each case in Figure 18. The top row (a) and (b) show the full modulated and compressed phase spaces of the two beams with the same vertical energy axes to assist comparison. As expected, the macroscopic shape of the two is very similar, with the key difference being the modulation wavelength and the CSR-induced energy loss from the final compression. In particular, the three-micron case has a larger magnitude of energy loss due to the stronger compressor. Nonetheless, each case reaches 4 kA peak current in the bunch train, which we highlight for a single microbunch in panels (c) and (d). The qualitative features are again similar, with each case having a peak current close to 4 kA and a relative slice energy spread around $3 \times 10^{-4}$ in the current spike. The one- and three-micron slices have a full width at half maximum of 44 nm and 119 nm, respectively, for this chosen central microbunch. Furthermore, the slice emittance in the microbunch has a value of 51 nm-rad and 66 nm-rad at the location of peak current, respectively. The projected normalized emittances (calculated over the microbunch FWHM) are 52 nm rad and 76 nm-rad, respectively. These emittances are consistent with the optimized FEL design demands, as we will see below. Furthermore, previous studies found the 1D CSR model used to overestimate the CSR-induced emittance growth in such microbunched beams [72], so even better final emittances may be expected from simulations employing more robust models.

## 8.1. Very-High-Frequency RF Devices for Bunch Compression

In order to perform the needed longitudinal phase-space linearization in the first compression stage, one needs a high-gradient RF accelerating structure with an integrated voltage of at least 15 MV working on the sixth harmonic of the main linac frequency (34.4 GHz). To power such a device, a multi-MW RF source in Ka-band must be developed. In this context, an electron gun with a lower perveance to obtain a high-efficiency klystron for driving phase-space-linearizing devices was investigated [73].

The proposed design allowed the development of a high-beam-quality electron gun dedicated to high-efficiency klystrons which are suitable for the next generation of multi-harmonic klystrons. Self-consistent analytic and numeric design for a set of electron guns with a high beam quality to be used in high-power Ka-band klystrons is presented [74]. In order to optimize the Ka-band klystron's efficiency for achieving 20 MW RF output power, different electron guns, beam-focusing channel designs, and RF beam dynamics have been examined and discussed. The electron flow is generated from a high-voltage DC gun (480 kV), and different beam currents (50 A, 100 A, and 218 A) have been extracted by changing the cathode–anode geometry in order to adjust the electric field equipotential lines. The electron beam is transported through the klystron channel, and the beam confinement is obtained by means of a high magnetic field produced by superconducting coils. The channel has been optimized to deliver high-power electron beams with a spot size of 2 mm diameter. As a result, an electron gun was chosen with 100 A, 480 kV operation in order to obtain an output RF power of about 20 MW and mitigate diamagnetic effects. With a SLED pulse compression system (Stanford Linac Energy Doubler) [75,76], one may obtain 80 MW output power. We note that this power source may energize a

phase-space linearizer and also a high-resolution (tens of femtosecond) RF sweeper-based diagnostic system.

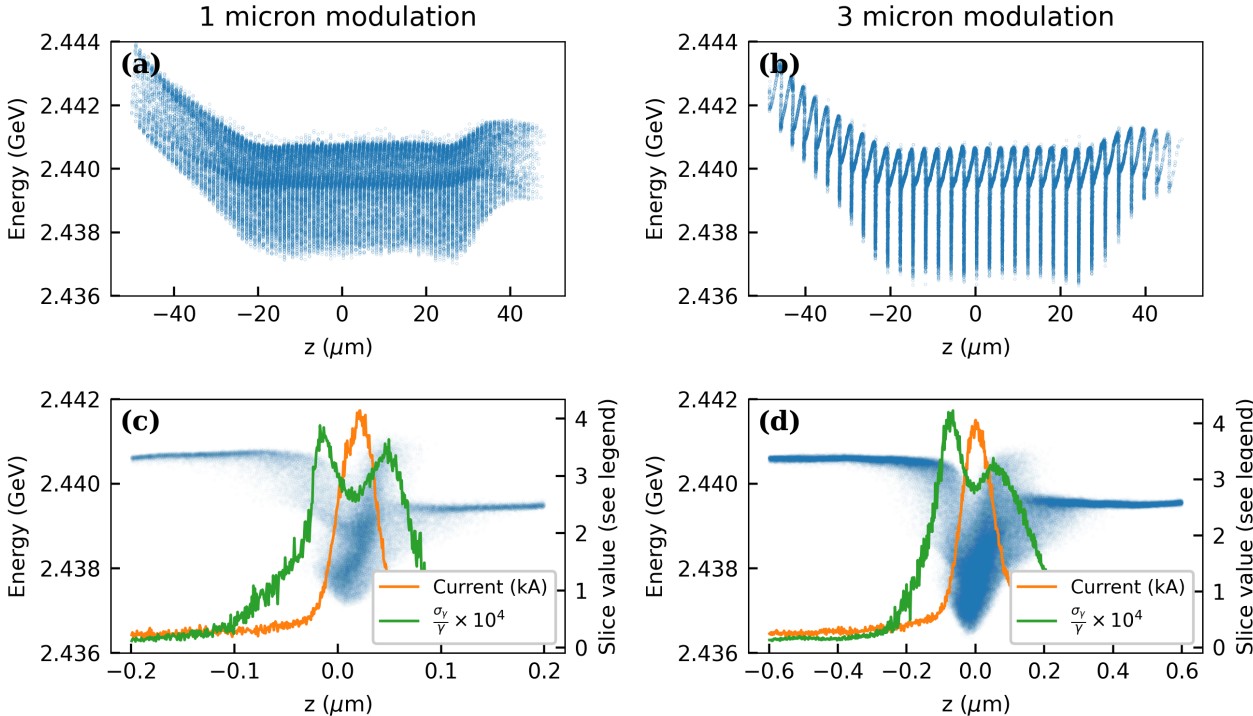

**Figure 18.** Characteristics of the longitudinal phase space of the electron beam at the end of the linac. (**a**,**b**) show the full longitudinal phase space, while (**c**,**d**) highlight the central microbunch, showing its current and slice energy spread profiles. The current and slice energy spread values numerically correspond to the axial markings on the right side of the plot. Plots (**a**,**c**) are for a one-micron laser modulation period, while (**b**,**d**) are for a three-micron period.

### 8.2. Long-Range BBU Effects after First Compression

After the first compression section, the bunches in the train are shortened by an order of magnitude, and their coupling to higher-order modes increases commensurately, with many modes excited. In the remaining part of this section we perform a multibunch BBU analysis with the code MILES that aims to evaluate the stability of the eight-bunch train in the main linac used for the XRAFEL scheme. The details of this multipulse configuration are dictated by the needs of the regenerative amplifier itself and are described in the following section. From a beam-stability perspective, the main concern of accelerating many bunches in the same RF macropulse is the parasitic excitation of the cavity higher-order modes (HOMs) introducing coupling among different bunches. In particular, monopole modes exchange energy with the beam, introducing bunch-to-bunch energy spread, whereas dipole modes induce transverse deflections responsible for potentially large oscillations of the bunches. The strength of the long-range wakefield (LRWF) interaction can be evaluated by the longitudinal and dipole impedances of the HOMs. The $R/Q$ values calculated with CST [77] are shown in Figure 19 for the $3\pi/4$ structure in comparison with the baseline $\pi$-mode design. The three possible irises $a = 1, 2, 3$ mm, corresponding to slight geometric variations of the nominal cavity, are investigated in order to make available detuning schemes for the mitigation of the BBU effects [78,79]. We note that the modes found in the $3\pi/4$ structure possess higher impedances, and thus, a stronger parasitic interaction is expected.

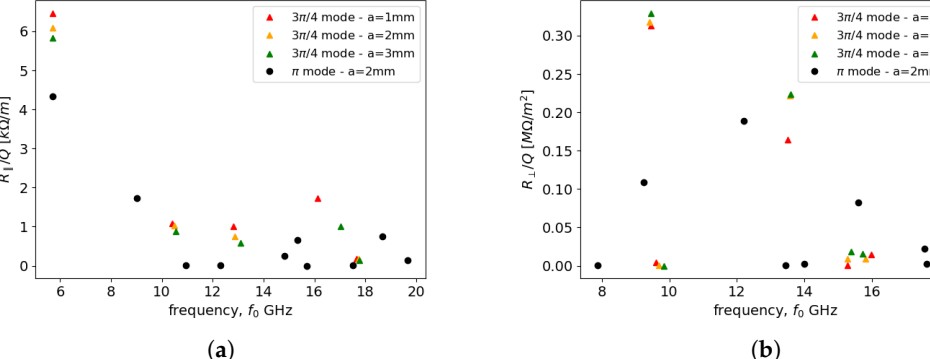

**Figure 19.** Monopole and dipole R/Q values versus frequency for a subset of the HOMs. The values obtained for the $3\pi/4$ structures are shown for different iris radii and compared with the baseline $\pi$-mode design. (**a**) Monopole modes; (**b**) dipole modes.

In the example that follows, we examine the LRWF effects in the main linac following the first-bunch compression stage. As mentioned above, this linac accelerates the electrons from 400 MeV to 2.44 GeV in 20 $\sim$1 m long sections operating at 100 MeV/m. Following the XRAFEL concept discussed in Section 9, we consider eight consecutive bunches, 100 pC each, with $\sim$40 ns separation (228 RF buckets) distributed over the 300 ns flat-top RF pulse. In Figure 20, we consider an alignment error of 100 μm so that the eight bunches are injected in the linac with an offset. The propagation off-axis causes excitation of the dipole HOMs with further deflections of the trailing bunches. Figure 20a shows the trajectories of the bunches in the linac, whereas Figure 20b shows the amplitude of the transverse oscillation envelope for all the bunches at the end of the linac as well as their energy. The transverse oscillations are normalized with respect to the amplitude obtained for the first bunch, which by causality is not subjected to BBU interaction, while the energies are expressed as deviations from the first-bunch energy ($\sim$2.44 GeV). The small bunch-to-bunch deviations show that LRWF effects are relatively weak due to the small number of pulses, their large time separation, and the moderate length of the linac that limits the mutual interaction. It is useful to remember that correction schemes capable of mitigating such deviations can be introduced. In particular, HOM dampers, as discussed in Section 6, or frequency detuning allow the suppression of the long range BBU interaction, whereas a proper time-dependent shaping of the RF macropulse can reduce the beam-loading effects.

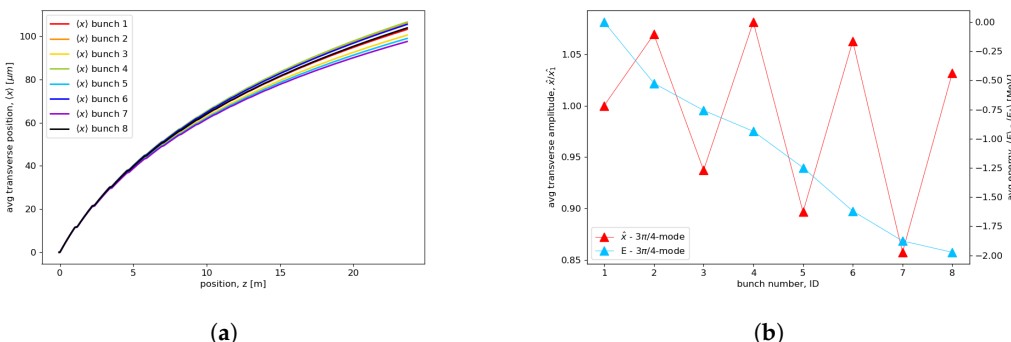

**Figure 20.** Trajectories and final amplitude of the beam oscillations for the eight bunches. (**a**) Bunch trajectories; (**b**) final bunch offset and energy.

### 8.3. Undulator Resistive Wall Wakefield Mitigation

A final issue concerning wakefields exists—the resistive wall effects inside the narrow-gap undulator. The IFEL bunch train current profile also serves to mitigate the undulator resistive wall wakefields (URWWs), which are an important consideration arising from the use of these short-period, small-gap undulators [80,81]. Ref. [82] discusses a regime of UR-

WWs particularly relevant to the present parameter space: a cryogenic, flat-wall geometry with ultrashort beams. In Figure 21, we analyze an idealized train of IFEL microbunches near the center of the macrobunch. These microbunches, spaced on 3 μm centers, pass through a cryogenic (RRR = 100) copper flat-wall structure with a gap size of 2 mm and an overall length of four meters. We compare this result to that of a single Gaussian beam with the same charge and peak current as the microbunched beam. The maximum induced energy spread due to the resistive wall wakes is seen to be less than 0.01 percent. This is suppressed by a factor of 15 compared to the case (also shown) of the full beam compressed to the 4 kA peak current.

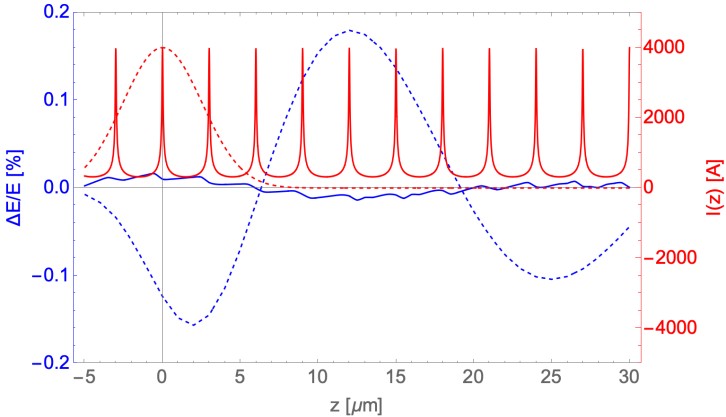

**Figure 21.** The relative energy loss for a central region of the microbunch train (solid) and a single bunch (dashed) as they transit a cryogenic (RRR = 100) copper flat-wall structure with a gap size of 2 mm and an overall length of four meters. Both beams have the same total charge peak current, but the train splits the charge over a series microbunches with 3 μm interbunch spacing. The relative energy loss in percent is shown in blue and the beam current in Amperes is shown in red.

## 9. X-ray Regenerative Amplifier FEL Performance

In this section, we consider an X-ray regenerative amplifier FEL (XRAFEL) design [83] with a multibunch C-band linac and present first simulation studies to obtain an estimate of the performance of the scheme. An XRAFEL wraps a high-gain FEL within a Bragg crystal cavity. The crystal cavity will monochromatize and return X-ray pulses for repetitive interactions with a bunch train generated by the C-band linac. The stable functioning of this scheme for storing hard X-rays has been shown [84].

The outcoupling of the XRAFEL radiation from the cavity is accomplished based on a recently developed scheme [50] termed the electron-beam-based *Q*-switching scheme. It relies on the introduction of an initial energy. During the amplification process, the beam is compressed by the undulator dispersion. This leads to a blueshift of the FEL microbunching wavelength, and part of the intracavity radiation spectrum is purposefully shifted out of the narrow reflection bandwidth of the Bragg mirror and is therefore transmitted. The remaining radiation is recirculated to seed the next lasing electron bunch.

The application of this technique to our current scenario was studied in [50], and further details beyond the present discussion are given there. In the following studies based on the methods introduced in Ref. [50], we consider eight electron bunches (each separated by 40 ns) that can be supported with a relatively short RF pulse of 300 ns. As a representative example, we consider a rectangular cavity composed of four diamond mirrors oriented at 45 degrees (see Figure 22), with Bragg resonance centered at 6.95 keV (Miller indices 220). The crystal thickness of 100 μm results in a peak reflectivity of 99.6% and an FWHM reflective bandwidth of 141 meV. The 4 m long undulators are centered between two mirrors M4 and M1, spaced 5.5 m apart in a rectangular X-ray cavity with round-trip length $L_c = 12$ m. These undulators are of a similar design to that employed in the UCXFEL study, as discussed above. Two compound refractive lenses (with 3 m focal length) are used

to control the transverse radiation size in the cavity. This focal length is optimized, giving consideration to the optical effects due to the significant gain in the undulator.

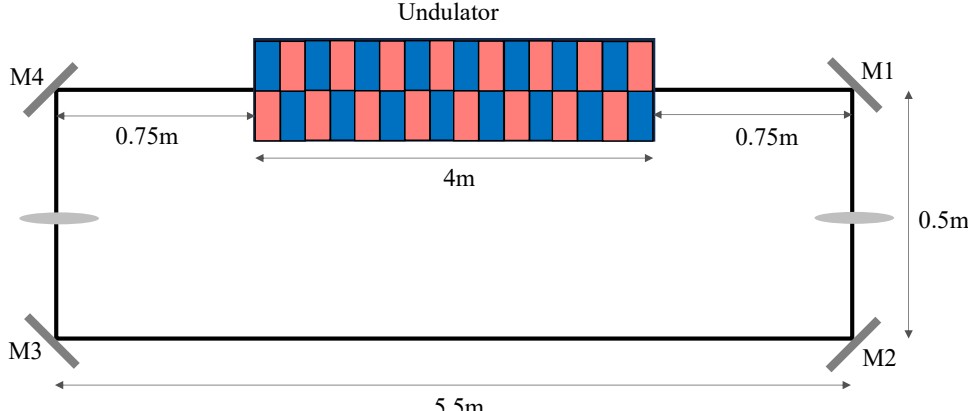

**Figure 22.** The setup of the 12 m round-trip cavity in an ultracompact XRAFEL. The radiation produced by lasing in the undulator is recirculated into the cavity and reflected via the Bragg condition for the (220) plane of diamond. The undulator is placed into one of the 5.5 m segments of the cavity. Lenses are used to control the transverse beam size of the radiation in the cavity, each having a 3 m focal length.

As shown in Table 2, a short-period (6.5 mm, with a 2 mm gap), cryogenically cooled permanent magnet undulator is considered for this design [85], as in the original UCXFEL study. This period permits operation at 2.44 GeV while obtaining 7 keV FEL photon energy. This approach necessitates the use of a very high 6D brightness beam. Further, in order to take advantage of the ultralow-emittance electron source, we insert a quadrupole FODO lattice superimposed on the undulator to provide the necessary focusing. This likely necessitates the use of permanent magnets placed inside the undulator gap. This design may be made more robust by using a new modified Panofsky quadrupole scheme [86].

**Table 2.** Summary of parameters for the ultracompact X-ray regenerative amplifier free-electron laser.

| Parameter | Units | Value |
|---|---|---|
| Energy | GeV | 2.44 |
| Energy spread | % | 0.03 |
| Normalized transverse emittance | nm-rad | 75 |
| Peak current | kA | 4.0 |
| Undulator parameter, K | | 0.501 |
| Undulator period | mm | 6.5 |
| Undulator length | m | 4.0 |
| Fundamental FEL wavelength | Å | 1.783 |
| Photon energy | keV | 6.95 |
| Diamond (220) bandwidth | meV | 141 |
| Cavity round-trip length (time) | m (ns) | 12 (40) |
| Number of electron bunches in an RF pulse | | 8 |

For the electron beam configurations, we considered both a 1 μm and 3 μm laser modulation period based on the bunch compression studies above. In the following GENESIS simulations, we consider a core part of the bunch of 20 μm in length. Both types of current modulations yield a 4 kA peak current; hence, the width of the 3 μm modulated spikes will be three times that of the 1 μm modulation. The slippage length for a 4 m undulator at 1.78 Å is about 100 nm. The longer current spikes (FWHM ∼120 nm) of the 3 μm current modulation are notably longer than the slippage length and hence have better FEL performance, both in power output and in spectral width. Detailed simulation studies

are discussed in another article of the same proceedings [50]; here, we only report the 3 μm current modulation SASE and XRAFEL results.

Figures 23 and 24 depict the GENESIS 1.3 results for the SASE and XRAFEL simulations using the 3 μm modulated beam profile in Figure 18b but with only a 20 μm flat bunch core to save the computation time. A single-pass SASE for a 3 μm modulated beam can reach a power of up to 6.5 GW with a noisy temporal profile and spectrum. By allowing the Bragg reflected X-rays to recirculate in a 12 m round-trip cavity and interact with the electron bunches multiple times, the maximum power output increases to 44 GW with a train of phase-locked pulses and a coherent spectrum (of multiple spikes). Steady-state XRAFEL behavior is reached in about six passes, by which point it is apparent that the XRAFEL design produces a higher power output, as well as a brighter spectral output by nearly two orders of magnitude. Thus, an XRAFEL can significantly enhance the temporal coherence and spectral brightness of the SASE FEL examined in Ref. [16] without notably increasing the footprint.

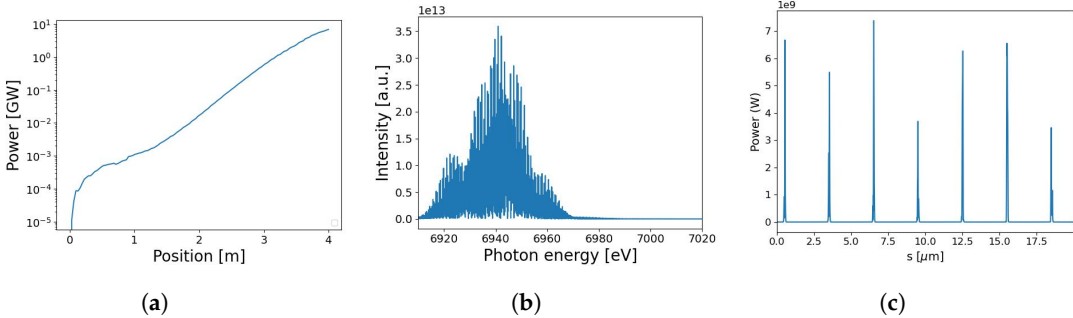

|          |          |          |
| :------: | :------: | :------: |
|   (**a**)  |   (**b**)  |   (**c**)  |

**Figure 23.** Results for single-pass SASE with no tapering for the 3 μm modulated beam. Each plot shows (**a**) the gain curve, (**b**) the output spectrum, and (**c**) the power profile. The maximum output power along the bunch peaks at 6.5 GW.

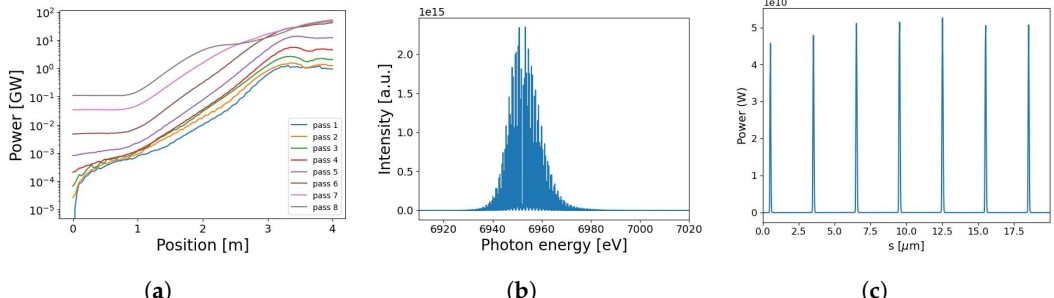

|          |          |          |
| :------: | :------: | :------: |
|   (**a**)  |   (**b**)  |   (**c**)  |

**Figure 24.** The (**a**) gain curve, (**b**) output spectrum, and (**c**) power profile for 3D XRAFEL with optimal tapering. A maximum output power (after exiting M1) of 44 GW is reached after the 6th pass.

We note that the XRAFEL using a current-modulated beam generates a train of phase-locked X-ray intensity spikes with a corresponding set of spectral spikes; while this temporal format can be further explored for various applications needing the highest spectral brightness, a more uniform electron bunch profile will generate a more compact X-ray temporal profile. In this way, the spectral brightness will be more concentrated in a single spike with the maximum coherence and brightness. In the near future, we plan to explore different bunch compression schemes to generate a compact single-pulse current distribution while mitigating deleterious collective effects. This will serve to further optimize the spectral brightness. This is discussed further below.

The XRAFEL is an example of a burgeoning new class of cavity-based X-ray free-electron lasers. This subfield has seen intense recent activity in both the theoretical [87] and experimental [84] arenas discussed here [88]. The XRAFEL itself has a high priority, as there is significant motivation in the field to search for XFEL solutions with very high peak and average-power X-ray, having enhanced coherence.

## 10. Conclusions and Outlook for Chip Metrology XFEL Development

This paper presented a conceptual design for a high average-flux UCXFEL at 7 keV photon energy using a high-repetition-rate system that utilizes a regenerative amplifier approach to the FEL. The departure point for this design effort was the UCXFEL design for 1 nm operation. In order to permit the needed increase in X-ray energy, flux, and coherence while presenting the most likely low-risk path to near-term realization, various changes in the soft X-ray design were examined. These include the use of a new type of C-band RF linac structure with efficient, streamlined power distribution and accelerating capabilities. At present, with a slightly derated gradient, we may consider the use of this linac operated at room temperature. We performed analyses indicating the likely compatibility of this new design with the preservation of beam quality due to various effects, prominently including space charge, coherent synchrotron radiation, and both long- and short-range wakefields.

This XRAFEL permits improvement in the average coherent flux available for ptychographic laminography over previous pioneering SLS experiments using the cSAXS beamline by five orders of magnitude. It is important in this case to quantify the spectral brightness expected, and in this case, we have seen that using a microbunching approach to final compression leads to a final spectral width of $2 \times 10^{-3}$, or an order of magnitude higher than that obtained at cSAXS [46]. This spectral width may have a detrimental effect on the ptychography measurements, and this must be studied. One may mitigate this added width by revisiting the compression scheme used. One must then consider collective effects such as the coherent synchrotron radiation and the resistive wall effect in the undulator.

To fully utilize higher flux for reaching higher imaging rates and resolution in an XFEL pulse format, there will need to be changes and improvements made in the PyXL measurement. With large, high-peak-intensity flux, it is possible to transversely split the XFEL pulse (in this case a train of eight pulses lasting 300 nsec) and perform a single-shot extended ptychography measurement as a component of the full scan. This approach, recently demonstrated at an XFEL [89], would ease the development challenges for faster and more accurate scanning techniques as well as aid in the mitigation of potential radiation damage. It should be noted that improvements in mechanical and electronic hardware are already capable of improving the PyXFL imaging rate by decreasing imaging and processing overheads [90]. In this regard, we note that the difference between the pulse structure envisioned for the UCXFEL discussed here is conducive for employing continuous sample motion in scans. To profit from the faster measurements permitted by higher flux in obtaining higher resolution, strategies are needed to avoid depth-of-field limitations, which should be lowered from 12 nm to 2 or 3 nm. Efforts are presently continuing to address this challenge, with notable progress achieved [91]. Ptychographic reconstruction techniques must be improved to reliably overcome the depth-of-field limitation on resolution, which is 12 nm for the present sample, and photon energy in order to reach 2 nm. In this regard, ptychography reconstruction algorithms are advancing rapidly, with a recent machine learning-based method published that demonstrates a much increased effectiveness in time and in demand for photons [51].

As a first step to proceeding towards development, we are developing a proposal for industrial development through US government sources. This proposal is informed by both the current paper and the original UCXFEL paper. It will address the issues needed to take this conceptual design study and mature it to full technical design. The main points to be addressed this next phase are as follows:

- Engage with imaging community and industrial users to set photon flux and spectral quality demands for ptychographic laminography chip inspection;
- Examine advantages of cryo-emission through measurements at emerging UCLA infrastructure;
- Finalize high-brightness RF gun technical approach based on cryo-RF experimental results;

- Fabricate new C-band linear accelerator sections with optimized power distribution requiring quadrant symmetry;
- Perform high-gradient testing of two-cell structures and one-meter linac sections at room temperature to achieve 100 MeV/m gradients;
- Perform cost optimization of RF and linear accelerator systems based on preliminary engineering design to determine technical choice for industrialized instrument;
- Optimize alignment and mechanical stability systems as well as active correctors for orbits;
- Consider beamline layout and design, including transport magnet systems, vacuum systems, and advanced transverse and longitudinal beam diagnostic systems;
- Quantify pulse–pulse variation beam quality, stability, and reproducibility as is critically important for image reconstruction;
- Reexamine XRAFEL design for increased spectral brightness and consistency with optimized ptychographic application;
- Design short-period cryogenic undulator with strong focusing and beam/radiation diagnostic systems;
- Consider X-ray optics design and engineering development for both XRAFEL and ptychography systems;
- Industrialize accelerator and FEL technical approach, including RF, cryogenic systems. This is in progress at RadiaBeam in the context of the present collaboration;
- Address the role of end station and detector technology. Consider the design, development, and integration of the ptychographic laminography system for industrial-scale chip metrology with collaborators from imaging community and the semiconductor industry;
- Integrate algorithm and big data challenges from ptychographic laminography into system design;
- Perform preliminary cost analysis of developing a prototype XRAFEL for chip metrology, with fully capable end station for fast inspection in the industrial environment.

This current effort at advanced applications of compact XFELs, reflected in the discussions above, is based on years of investment by federal agencies (DOE, DARPA, NSF). We propose here to exploit these results to transform the capabilities of the microelectronics fabrication industry. This finds particular urgency in light of the stated goals of metrology in the context of the CHIPS Act of 2021. In this regard, it should be noted that many of the capabilities introduced in the design of the compact XRAFEL discussed here may also be employed for the metrology of lithography masks. These issues are examined in some detail in the review found in [92]. Ptychographic techniques using Angstrom-class X-rays, enabled by a UCXFEL similar to that described here, may provide a key tool for this type of metrology as well. The list of potential applications of the concepts we introduced in this article thus seems likely to expand further.

**Author Contributions:** This article is the product of a wide collaborative effort, conceived and integrated by the lead author, J.B.R. The effort is broken down as follows: cryogenic RF research at the UCLA MOTHRA Lab was performed and written up by G.L. and S.O.; G.A., Y.S., P.M. (Pratik Manwani), P.M. (Pietro Musumeci), M.Y. and O.W. contributed text on UCLA UCXFEL development; F.B., M.M. and L.P. produced the analysis of beam breakup; M.C., E.C., A.F., P.M.A., J.M. (Jared Maxson) and F.B. performed and summarized the low energy beam dynamics analysis; R.R. led and summarized the study of beam compression dynamics; L.G., J.M. (Janwei Miao) and J.B.R. evaluated the implementation of ptychographic laminography in the UCXFEL design; advanced RF device development research was led by S.T. and performed by Z.L., B.S., A.M. (Andrea Mostacci) and E.N.; novel RF device development and testing at LANL was performed and summarized by E.I.S., H.X. and D.K.; industrial development of RF and magnetic systems for the UCXFEL at RadiaBeam was performed and summarized by A.M. (Alex Murokh), A.A., R.A. and S.K.; evaluation and description of resistive wall wakefield effects in the undulator was performed by N.M.; the XRAFEL analysis was performed and described by Z.H., M.S. and J.T. All authors have read and agreed to the published version of the manuscript.

**Funding:** This work was performed with support of the National Science Foundation, through the Center for Bright Beams, Grant No. PHY-1549132; through an NSF Science and Technology Center. Support was also obtained from the US Dept. of Energy, Division of High Energy Physics, under contracts no. DE-SC0009914, no. DE-SC0020409, and DOE/SU Contract No. DE-AC02-76-SF00515. Additional support has been given by through the DARPA GRIT program under Contract No. HR001120C0072, and the Los Alamos National Laboratory (LANL) LDRD Program.

**Data Availability Statement:** The raw data supporting the conclusions of this article will be made available by the authors on request.

**Acknowledgments:** Author James Rosenzweig would like to acknowledge useful conversations with A.F.J. Levi (Univ. of Southern California).

**Conflicts of Interest:** The authors declare no conflicts of interest.

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
