# Peer review of "A High-Flux Compact X-ray Free-Electron Laser for Next-Generation Chip Metrology Needs"

_instruments, doi:10.3390/instruments8010019_

Round 1

Reviewer 1 Report

Comments and Suggestions for Authors

Comments on ’Development of a High-Flux Free Electron Laser for Advanced Semiconductor Chip Evaluation’

The paper addresses the development of a high-flux free electron laser (FEL) light source with potential applications in the evaluation of advanced semiconductor chips. The semiconductor industry's continuous progress has led to nanometer-scale elements in cutting-edge devices, necessitating evaluations with exceptionally high spatial resolution. Presently, thin-section materials from these devices are commonly prepared and assessed using transmission electron microscopes, but there is a growing demand for non-destructive 3D testing methods. This paper explores the practical application of X-ray free electron lasers as a solution to this challenge, despite the existence of other X-ray sources like diffraction-limited storage ring light sources using MBA technology.

The paper effectively communicates its objectives and motivations, which center around the development of a high-flux free electron laser for the practical application of evaluating advanced semiconductor chips with high spatial resolution. The introduction provides clear context for the study and its relevance to industrial needs.

The paper succinctly outlines the current state of semiconductor technology and the challenges posed by nanometer-sized elements, highlighting the importance of non-destructive 3D testing methods. The mention of ptychographic-laminography as an existing method and the need for an improved X-ray source is well-founded.

The language used throughout the paper is clear and concise, enhancing the reader's understanding of the subject matter. The writing is professional and free from grammatical errors.

The paper concludes by emphasizing its focus on the practical application of X-ray free electron lasers as a potential solution for high-resolution semiconductor chip evaluation. It successfully conveys the significance of this research direction and its potential impact on the semiconductor industry.

Overall, the paper is well-structured, informative, and effectively addresses its research objectives. It provides a valuable contribution to the field of semiconductor chip evaluation and the development of advanced X-ray sources. I recommend it for publication with minor revisions, such as further elaboration on the advantages and challenges associated with X-ray free electron lasers and their potential for industrial use. If possible, please mention the usefulness of XFEL as an EUV light source for semiconductor chip manufacturing.

Comments on the Quality of English Language

The quality of English is sufficient. Please fix some typos.

Author Response

We are gratified by the understanding shown of our work by Referee 1, and kind endorsement given. This is a long and involved work, and the review is much appreciate. Concerning the concrete suggestion to mention the application to FELs in other industrial contexts, we have included further discussion of the application of soft X-rays FEL sources to chip mask lithography, and hard X-rays to inspection of lithography masks. We also provide references to other industrial application of X-ray FELs in industrial nanomaterials.

Typos and other minor expostion problems have been addressed (see response to Referee 2 for further details)

Reviewer 2 Report

Comments and Suggestions for Authors

In general, the paper describes a highly interesting and revolutionary design which – if it could be realized experimentally – might be fundamentally reformative to the chip industry and therefore decisive on a societal level. Therefore, the topic merits publication in general.

However, firstly, there are very many English language problems in this manuscript. They range from simple regrettable typos, spelling errors and word duplications to more subtle grammatical and style mistakes.

The interesting topic of the manuscript is unfortunately much damaged by these deficiencies.

Here is a (non-exhaustive) selection:

line 1: a ultra-compact ïƒ  an ultra-compact

line 1: ultra-compact X-ray free-electron laser ïƒ  X-ray 1 free-electron laser (UCXFEL) 
           Note: introduce the abbreviations UCXFEL and XFEL in the abstract

line 9: to addressed ïƒ  to address

line 15: “extending the UCXFEL to harder X-rays (7 keV)”: extending from where? Was there a previous study? Some context is apparently missing here.

line 23: “of of” ïƒ  of

line 28: “coherence” ïƒ  transverse coherence

line 50: “with single-shot CDI-based on XFELs” is not a valid expression

line 89: delete superfluous “the”

line 105: “one may impact” is logically not good English in this context

line 108: replace “frontier resolution” either with a commonly accepted expression or an actual numerical value for resolution

line 109: “system has a clear vision”: a system cannot have a vision.

line 134: “by the lowered the beam energy needed” is incorrect English

line 164: “bright beam in hand” : you cannot have this beam “in the hand”, thus this is an unfortunate expression. 

line 164: what is “a 1 nm design”? You probably mean a design for a 1 nm wavelength radiation source?

line 212: the expression “challenges - and opportunities - are presented in this experimental scenario spanning the gamut of technological development and basic physics.” is very flowerish but unnecessarily unscientific, not specific, and doesn’t add content to the paper.

Figure 4 legend: “We expect obtain factor of” ïƒ  “We expect to obtain a factor of” 

Section 2 has a subsection 2.1 but no other subsection. It seems like some sub-sectioning is missing here. The phrase in line 195/196 “development on the cryo-RF gun, cryogenic photoemission and general C-band (4-8 GHz) techniques, and wakefield management” implies that there would be more subsections than only the one on “Cryogenic photoinjector development”.

line 292: What is “CHIPS Act-oriented research”? It is not explained.

line 339: “well-sub-hour” is not an English expression

line 342: “is based on low energy pre-bunched beams is obtained” contains two verbs, no valid English sentence

line 361: “preformed” ïƒ  performed?

line 496 “the piece is fabricated” ïƒ  which is fabricated?

line 532: “nickel and Chrome”: elements are either upper case when abbreviated or lower case as full word, but certainly not both upper and lower case

line 677: “is as was reported”: incomplete sentence structure

Figure 21 caption: “as a percent” ïƒ  in percent. “The energy loss as a percent is in blue and the beam current is in red.” ïƒ  relative energy loss expressed in percent and beam current in Ampere, shown in red.

Figure 22 caption: “The setup of the 12 meter round-trip cavity an ultra-compact XRAFEL”. Either the “an” is wrong or a word is missing.

line 891: “The role ole of end station” – obvious partial word doubling

line 899: “XFELS” is a wrong abbreviation ïƒ  “XFELs”

The following are more or less style issues which inhibit better comprehension of the text:

line 375: “multi-pulse mode” can mean a lot of different things. This expression was not introduced in the text before line 375, therefore it is entirely unclear what is meant by this.

line 378: “multiple beam bunches” probably refers to “multiple electron bunches”? Make it explicit!

line 492: “In order to proceed with a design aimed at near term implementation of a chips- 492 metrology UCXFEL…” It is not necessary to repeat this over and over in every section. It is really appalling to read and disrespectful to the avid reader who probably understood this the first time.

Problems with figures:

·       Figure 8: the text inside this figure has unacceptably low resolution. It is blurry.

·       Figure 10: same problem as for figure 8 with blurry fonts on the axes of the plots.

·       The resolution of the text in both figures doesn’t match the resolution of the text in most of the other plots.

·       Figure 11 does not show clearly what is written into the figure caption, and descriptions inside the graph are missing which would point to the various points of interest

·       Figures 18 (c) and (d) show a trace called current (kA) in orange, but there is no y-axis for the numerical values of this curve. The green curve values seem to be shown on the right y-axis and the blue data on the left y-axis, but also this is not very clearly indicated.

In terms of content, there are also issues:

·       line 136: formula ρ ∝ γdirectly contradicts with formula (2) which states ρ ∝ γ6

·       line 330-339: the argumentation only expresses the increase in brightness, but lacks the description how this directly translates into reduction of required experimental beamtime duration.

·       line 510/511: “the 4-fold symmetry shown in Fig. 13” – there is no 4-fold symmetry in Fig. 13

·       line 795: the distance between M1 and M2 is not 5.5m as can be seen from Figure 22, but it is 0.5m Inconsistent.

·       Figure 22 caption: here, the radiation is produced by the undulator shown in the figure, not “produced by the XFEL” as written in the caption.

·       Figure 22: why 3m focal length? Why not 3.25m as would be obvious from the geometry? If this difference is of any importance, why is it not explained?

·       Chapter 8 doesn’t mention at all the entire challenge of coupling out the produced photons after creating them inside the cavity.

·       It should be mentioned more clearly that the XRAFEL concept in itself has not yet been demonstrated at all experimentally for hard x-rays, let alone for such a revolutionary different linear accelerator.

·       In line 850-859 the comparison between a synchrotron beamline and the new concept performance is made stating huge improvements, but in fact it should rather be compared to performance of XFEL beamlines.

·       Overall, it is not explained if one of the key features of XFEL radiation, namely the ultrashort pulse duration in the femtosecond range, is of any importance for this concept.

Authors Contributions

The Section on Authors Contributions is not detailed enough at all. “This article is the product of a wide collaborative effort by of all the authors listed.” There should be at least some indication on for example who contributed to which sections or topics, because certainly not everyone contributed to all these various fields of physics.

Comments on the Quality of English Language

there are very many English language problems in this manuscript. They range from simple regrettable typos, spelling errors and word duplications to more subtle grammatical and style mistakes.

The interesting topic of the manuscript is unfortunately much damaged by these deficiencies.

Here is a (non-exhaustive) selection:

line 1: a ultra-compact ïƒ  an ultra-compact

line 1: ultra-compact X-ray free-electron laser ïƒ  X-ray 1 free-electron laser (UCXFEL) 
           Note: introduce the abbreviations UCXFEL and XFEL in the abstract

line 9: to addressed ïƒ  to address

line 15: “extending the UCXFEL to harder X-rays (7 keV)”: extending from where? Was there a previous study? Some context is apparently missing here.

line 23: “of of” ïƒ  of

line 28: “coherence” ïƒ  transverse coherence

line 50: “with single-shot CDI-based on XFELs” is not a valid expression

line 89: delete superfluous “the”

line 105: “one may impact” is logically not good English in this context

line 108: replace “frontier resolution” either with a commonly accepted expression or an actual numerical value for resolution

line 109: “system has a clear vision”: a system cannot have a vision.

line 134: “by the lowered the beam energy needed” is incorrect English

line 164: “bright beam in hand” : you cannot have this beam “in the hand”, thus this is an unfortunate expression. 

line 164: what is “a 1 nm design”? You probably mean a design for a 1 nm wavelength radiation source?

line 212: the expression “challenges - and opportunities - are presented in this experimental scenario spanning the gamut of technological development and basic physics.” is very flowerish but unnecessarily unscientific, not specific, and doesn’t add content to the paper.

Figure 4 legend: “We expect obtain factor of” ïƒ  “We expect to obtain a factor of” 

Section 2 has a subsection 2.1 but no other subsection. It seems like some sub-sectioning is missing here. The phrase in line 195/196 “development on the cryo-RF gun, cryogenic photoemission and general C-band (4-8 GHz) techniques, and wakefield management” implies that there would be more subsections than only the one on “Cryogenic photoinjector development”.

line 292: What is “CHIPS Act-oriented research”? It is not explained.

line 339: “well-sub-hour” is not an English expression

line 342: “is based on low energy pre-bunched beams is obtained” contains two verbs, no valid English sentence

line 361: “preformed” ïƒ  performed?

line 496 “the piece is fabricated” ïƒ  which is fabricated?

line 532: “nickel and Chrome”: elements are either upper case when abbreviated or lower case as full word, but certainly not both upper and lower case

line 677: “is as was reported”: incomplete sentence structure

Figure 21 caption: “as a percent” ïƒ  in percent. “The energy loss as a percent is in blue and the beam current is in red.” ïƒ  relative energy loss expressed in percent and beam current in Ampere, shown in red.

Figure 22 caption: “The setup of the 12 meter round-trip cavity an ultra-compact XRAFEL”. Either the “an” is wrong or a word is missing.

line 891: “The role ole of end station” – obvious partial word doubling

line 899: “XFELS” is a wrong abbreviation ïƒ  “XFELs”

Author Response

We would like to express our gratitude to the referee (2) for a careful reading of the manuscript, and for their comment on the overall impact of the x

“In general, the paper describes a highly interesting and revolutionary design which – if it could be realized experimentally – might be fundamentally reformative to the chip industry and therefore decisive on a societal level. Therefore, the topic merits publication in general.” The other referee (1) is in strong agreement on this point, and so the authorship shares a sense of gratitude over this validation.

As this manuscript was created with some calendar considerations arising from its inclusion in a special issue of Instruments, it is of uncommon length, and represents the work of many authors, some of which are not native English speakers, it is inevitable that style, typographic and graphical problems be encountered. We appreciate the detailed advice on sorting through these issues, and believe we have comprehensively addressed them. In addition, there were a few some content-related comments that we have responded to with changes to the manuscript. We now believe that the clarity and impact of the paper are greatly improved through the need to react to the useful suggestions of the referees.

For referee 1 comments and suggestions which required simple correction or implementation, we do not response; all such points which are not mentioned have been addressed by simple adoption of the referee’s suggested changes.

For others, some complexity in our response demands a short explanation to the referee’s comment or suggestion. What follows is a list of such responses, given point-by-point with the original referee comment/suggestion:

-------------

line 15: “extending the UCXFEL to harder X-rays (7 keV)”: extending from where? Was there a previous study? Some context is apparently missing here. We had mentioned a few lines above that the original UCXFEL was analyzed for 1 nm operation. We have added the energy of the photon for added clarity.

line 212: the expression “challenges - and opportunities - are presented in this experimental scenario spanning the gamut of technological development and basic physics.” is very flowerish but unnecessarily unscientific, not specific, and doesn’t add content to the paper. This introductory sentence has been shortened and clarified.

Section 2 has a subsection 2.1 but no other subsection. It seems like some sub-sectioning is missing here. The phrase in line 195/196 “development on the cryo-RF gun, cryogenic photoemission and general C-band (4-8 GHz) techniques, and wakefield management” implies that there would be more subsections than only the one on “Cryogenic photoinjector development”. This subsection has been reclassified to be a section.

  • line 136: formula ρ γ4 directly contradicts with formula (2) which states ρ ∝ γ6. This was a typographical error, we are grateful to the referee for pointing this out. Formula 2 stands.
  • line 330-339: the argumentation only expresses the increase in brightness, but lacks the description how this directly translates into reduction of required experimental beamtime duration. We have added significant context from the literature, where this issue is quantitatively discussed. The ptychographic laminography community has identified the issues associated with use of significantly higher flux. We review their conclusions, providing the relevant references to current literation, and add our own comments and clarifications on recent progress.
  • line 510/511: “the 4-fold symmetry shown in Fig. 13” – there is no 4-fold symmetry in Fig. 13. This is corrected to “quadrant structure”
  • line 795: the distance between M1 and M2 is not 5.5m as can be seen from Figure 22, but it is 0.5m Inconsistent. The text has been corrected to indicate the distance between M4 and M1
  • Figure 22 caption: here, the radiation is produced by the undulator shown in the figure, not “produced by the XFEL” as written in the caption. We have written “produced by lasing in the undulator” for further clarity, as the radiation is not produced directly by the the undulator per se.
  • Figure 22: why 3m focal length? Why not 3.25m as would be obvious from the geometry? If this difference is of any importance, why is it not explained? We explain in an added sentence: “This focal length is optimized giving consideration to the optical effects due to the significant gain in the undulator.”
  • Chapter 8 doesn’t mention at all the entire challenge of coupling out the produced photons after creating them inside the cavity. This issue is described in detail in several references, including Singleton, et al., published in this special issue of Instruments. We review this scheme, which exploits the limited reflection bandwidth of the cavity mirrors, in combination with chirping of the beam pulse energies.
  • It should be mentioned more clearly that the XRAFEL concept in itselfhas not yet been demonstrated at all experimentally for hard x-rays, let alone for such a revolutionary different linear accelerator. We add significant references on the current state of development on cavity-based XFELs, giving theoretical resources and discussing recent experimental progress. We mention the lack of demonstration of the XRAFEL per se, while pointing out the considerable activity promoting the XRAFEL and the strong motivation for doing so – high average and peak power with narrow bandwidth.
  • In line 850-859 the comparison between a synchrotron beamline and the new concept performance is made stating huge improvements, but in fact it should rather be compared to performance of XFEL beamlines. We do not provide a direct comparison to XFEL beamlines for ptychographic laminography as they are currently not applicable to such measurements. This is due to the time involved in the PyXL approach, and the lack of opportunity to use XFELs for this application points out the need for the proposed UCXFEL. We now discuss these issues as well as the new aspects of the beamline relevant to the adoption of the XRAFEL pulse output – management of peak intensity, and effects on scanning.
  • Overall, it is not explained if one of the key features of XFEL radiation, namely the ultrashort pulse duration in the femtosecond range, is of any importance for this concept. We discuss in the manuscript the issues of peak power management in the optics, the utility of the short pulse in scanning, and the advantage associated with high peak power to enable tomography based on multiple XFEL spots used simultaneously.

 line 492: “In order to proceed with a design aimed at near term implementation of a chips- 492 metrology UCXFEL…” It is not necessary to repeat this over and over in every section. This has been rephrased and shortended to avoid repetitive statements.

Problems with figures: All addressed.

The authors’ contributions listed are now complete and detailed, per request.

Round 2

Reviewer 1 Report

Comments and Suggestions for Authors

The current referee believes that the manuscript holds significant value for publication, even in its pre-revised form. He has figured out that the manuscript's strengths have been further enhanced through the revision. Therefore, there is no objection to proceeding with the publication in its current state.

Author Response

We thank you for your kind endorsement of  this work. 

Reviewer 2 Report

Comments and Suggestions for Authors

The authors have addressed the reviewer comments and implemented changes as adequate. 

Comments on the Quality of English Language

The edited and new sentences should be checked once more because some new English language / spelling errors were introduced. Examples for such errors: line 194 ("transvere"), line 207 ("usch"), line 382 ("It and relates techniques"), line 383 ("currentlly") and so on. The sentence line 928-930 lacks a verb. There are some apparently broken references in the additional texts, see line 827 and line 933.

Author Response

Thank you for your patient reading. We have corrected all identified problems per your suggestion